

# Comparative larval morphology of four *Pteroptyx* (Coleoptera, Lampyridae, Luciolinae) species in Thailand

Suparada Boonloi[1,*], Parichart Laksanawimol[2,*], Soraya Jaikla[3,4], Marc A. Branham[5] and Anchana Thancharoen[1,4,6]

[1] Department of Entomology, Kasetsart University, Bangkok, Thailand
[2] Faculty of Science, Chandrakasem Rajabhat University, Bangkok, Thailand
[3] Department of Entomology, Kasetsart University, Kamphaeng Saen, Nakhon Pathom, Thailand
[4] IUCN Firefly Specialist Group, Gland, Switzerland
[5] Department of Entomology and Nematology, University of Florida, Gainesville, FL, United States of America
[6] Research and Lifelong Learning Center for Urban and Environmental Entomology, Kasetsart University Institute for Advanced Studies, Kasetsart University, Bangkok, Thailand
[*] These authors contributed equally to this work.

Corresponding author
Anchana Thancharoen,
koybio@gmail.com

## ABSTRACT

**Background**. Fireflies in the genus *Pteroptyx* are renowned for their significance in firefly tourism initiatives. Their occurrence and abundance have been extensively studied to facilitate sustainable utilization and conservation of their populations. As a group of highly charismatic insects, *Pteroptyx* fireflies play an important role in raising awareness and advocating for the conservation of mangrove forests. Previous taxonomic studies on these fireflies have primarily focused on adult while the larvae remain limited. Therefore, this study aimed to compare the larval morphology of four *Pteroptyx* species found in Thailand.

**Methods**. The characteristics of the larvae were examined under a microscope, and the species were identified by rearing the larvae to adult and comparing the adult males with previously identified male specimens. Additionally, morphometric analysis of the protergites was conducted to support identification efforts.

**Results**. Larval descriptions and an identification key for four *Pteroptyx* species were developed through the comparison of larval morphology, focusing on tergite texture, protergum shape, color patterns, mouthparts, and the holdfast organ (pygopod). Among the species, *P. valida* is uniquely characterized by lacking tubercles on the thoracic tergites and has a holdfast organ composed of more than 10 retractable filaments, clearly distinguishing it from the other three *Pteroptyx* species. Most species have a protergum length much longer than its width, except *P. tener*. An overview description of *Pteroptyx* larvae is provided with supplementary files summarizing the key characteristics of these four *Pteroptyx* larvae.

**Discussion**. Species-specific traits are evident among the four species, likely reflecting their specific biological and ecological requirements. *Pteroptyx valida* Olivier, 1909 displays distinct morphological characteristics, including features of the holdfast organ.

## INTRODUCTION

Fireflies in the genus *Pteroptyx* exhibit some of the most charismatic bioluminescent flashing behavior known to occur in the family Lampyridae. Large numbers of these beetles will congregate on specific trees in intertidal riparian mangrove habitats and flash synchronously (*Jaikla et al., 2020a*; *Jusoh, Hashim & Ibrahim, 2010*; *Prasertkul, 2018*; *Wong, 2022*). These mass aggregations and synchronous flashing behavior has made *Pteroptyx* fireflies an economically important insect in ecotourism in both Thailand and Malaysia (*Lewis et al., 2021*; *Thancharoen, 2012*). Firefly habitats are being destroyed and face numerous anthropogenic threats, resulting in an obvious decline in populations (*Idris et al., 2021*; *Lewis et al., 2020*; *Owens et al., 2022*; *Thancharoen & Masoh, 2019*). Recently, five species of *Pteroptyx* were assessed and included on the IUCN Red List with one categorized as endangered (*Yip & Yiu, 2023*) and four as vulnerable (*Nada et al., 2024a*; *Nada et al., 2024b*; *Jusoh et al., 2024*; *Thancharoen et al., 2024*). Habitat restoration and conservation programs have been recommended to conserve insect diversity and abundance (*Harvey et al., 2020*). Therefore, identifying larvae is the essential first step in initiating habitat conservation, which has been challenging in the past.

There are several limitations to conducting research on firefly larvae. For example, there are difficulties observing their faint light at night, insufficient identification keys, and challenges in associating unknown larvae with identified adult specimens, especially with many sympatric species. Larval-adult associations are needed to perform or confirm an accurate species identification of larvae, through the comparison of larvae to identified adult male specimens. This can be accomplished by rearing collected larvae to the adult stage or allowing collected or/and mated females to lay eggs which are then reared to the adult stage (*Archangelsky & Branham, 1998*; *Ballantyne & Menayah, 2002*; *Branham & Archangelsky, 2000*; *Fu & Ballantyne, 2024*; *Fu, Ballantyne & Lambkin, 2012b*; *Ohba & Sim, 1994*; *Rosa, 2007*; *Viviani, Rosa & Martins, 2012*; *Zurita-García et al., 2022*), as well as larval-adult associations accomplished using molecular methods (*Jusoh et al., 2014*; *Nada, Ballantyne & Jusoh, 2021*). However, rearing fireflies is very challenging and has only been done successfully in a few *Pteroptyx* species (*Ballantyne & Menayah, 2002*; *Jaikla, Thancharoen & Pinkaew, 2020b*; *Loomboot et al., 2007*).

The morphology of firefly larvae varies in relation to the ecological diversity of the species, with larvae found in aquatic, semiaquatic, and terrestrial habitats. Aquatic larvae exhibit unique features, such as abdominal gills or metapneustic traits, which are absent in the other groups (*Fu, Ballantyne & Lambkin, 2012b*). Terrestrial larvae were classified into two groups: those with laterally explanate tergal margins and those without. *Pteroptyx* larvae fall into the group lacking laterally explanate tergal margins. Only two of the 18 described species of *Pteroptyx* fireflies have known and described larvae, *P. valida* and *P. maipo* (*Ballantyne et al., 2011*; *Ballantyne et al., 2019*; *Ballantyne & Menayah, 2002*; *Jusoh et al., 2018*). Although the larval morphology of these two species was well described, knowledge of the larvae from the remaining *Pteroptyx* species is sorely needed. Through the rearing of collected larvae and observing egg-laying adults to confirm species identification, we evaluate and describe the external morphology of four *Pteroptyx* species found in Thailand.

**Table 1  Collection locations of fireflies with the numbers of collected specimens and reared larvae.**

| Species | Location | GPS | No. collected fireflies | | No. larvae reared |
|---------|----------|-----|--------|--------|------|
| | | | Adults | Larvae | |
| *P. asymmetria* | Krabi | 8°06′1.7″N, 98°57′52.71″E | 13 | 18 | 183 |
| *P. malaccae* | Samut Prakarn | 13°39′38.3″N,100°33′5.1″E | 26 | 30 | 116 |
| *P. tener* | Trang | 7°28′46.8″N,99°32′46.8″E | 27 | 7 | 268 |
| *P. valida* | Krabi | 8°05′40.6″N,98°54′59.63″E | 36 | 66 | 346 |

We compare the morphology of instar larvae and provide a larval identification key for *Pteroptyx* species currently known from Thailand.

## MATERIALS & METHODS

### Rearing

Adults and larvae of *Pteroptyx* spp. were collected from various riparian locations within mangrove areas in Thailand from 2023 to 2024. The research was approved for animal care and use for scientific research at Kasetsart University (ACKU67-AGR-022), and the corresponding author obtained an animal use license (U1022522558). Adults were collected from foliage using a sweep net and larvae were hand collected (Table 1). All larvae were found within the areas that contained the adult display trees. Adults were placed in circular transparent plastic boxes (four cm in diameter, six cm in height, with a one cm diameter aeration hole on the lid). A 30% honey solution on moist cotton served as food and moist cotton served as the oviposition and larval substrate. After females laid eggs on the moist cotton, the cotton with eggs was transferred to new containers to hatch at temperatures ranging from 26 to 30 °C under natural photoperiod conditions. During the egg stage, those that were unviable and started to rot were removed from the rearing containers. *Assiminea* (Gastropoda: Assimineidae) and *Ellobium* (Gastropoda: Ellobiidae) snails from the same habitat were provided as food throughout all larval stages. Contact with the fragile first instar larvae were avoided. The old cotton and consumed snails were removed every two days to prevent the growth of fungi, *etc*. The species identity of larvae was confirmed by both observing the development of collected larvae into adults, as adults can be identified, and observing the eggs laid by collected adults hatching into larvae. The adults were identified using the identification keys of *Ballantyne et al. (2015)* and *Jusoh et al. (2018)*.

### Morphological examination

The first instar larvae of all species were examined, while fifth instar were used for *Pteroptyx asymmetria* and *P. malaccae* and sixth instar for *P. tener* and *P. valida*. The use of different late instar larvae was due to the availability of rearing stock for each firefly species. Based on observations, the fourth to sixth larval instars of *Pteroptyx* share similar characteristics. Twelve larvae of each species were used for morphological study. To accurately document larval morphology, including soft structures, live larvae were promptly immersed in 70% ethanol before photography. The instars of the larvae were confirmed by molting

during rearing to study their life cycle. Photography was conducted using a Leica DFC 295 camera attached to a Leica S8 APO Stereozoomtrinocular microscope (Leica, Heerburgg, Switzerland). Image stacks were combined into a single image, and measurements of the most representative specimen in our study series were performed using Leica Application Suite V4.2. The resulting images were then edited using Photoshop CS8 (Version 6.1) (Adobe Inc.). Described specimens were preserved and deposited at Kasetsart University (KU), Thailand.

Three larvae of each species (a total of 12 individuals) were dissected to separate and examine their mouthparts. The live specimens were placed on ice for 10 to 15 min prior to preservation in 70% ethanol. This chilling process allowed the neck membrane to relax, and become more extendible, thereby projecting the head outside and forward of the pronotum. Antennae were separated from the head, and the remaining parts were divided into two sections: mandibles and the labium-maxillae complex. The hind legs were then detached from the body. All dissected pieces were slide-mounted on glass microscope slides using modified Berlese's medium (*Amrine & Manson, 1996*) before being photographed with a Leica DM 750 camera attached to a Leica ICC50W (Leica, Heerbrugg, Switzerland). Once slide-mounted, the sensory cone and the third antennal segment on the antennae and the setae on the metafemur were measured.

Numbers of retractable filaments on the holdfast organ were videotaped, photographed, and counted while placing the live larvae in water depths ranging from three to five mm using a Leica EZ4 W stereo microscope (Leica, Wetzlar, Germany).

## Morphological measurements

The measurements of larval morphological characters were carried out on live specimens in order to facilitate further biological studies; therefore, some errors in the measurements of body length and body width may have occurred. However, the specimens were chilled on ice for 5–10 min to reduce activity and relax muscle tension. The larvae subjected to the ice treatment displayed a more pronounced horizontal alignment of prothorax compared to those that did not undergo treatment (Fig. S1), resulting in more accurate measurements.

Measurements of body length, body width, protergum length (PL), width (PW), and distance between each tubercle (t1–t5) (Fig. S2) were conducted on 41 larvae of four *Pteroptyx* species (10, 11, 9 and 11 larvae for PA, PM, PT, and PV, respectively) using a Leica EZ4 W stereo microscope (Leica). The ratio of PL and PW was calculated for morphological comparison. The t1–t5 difference within each species was analyzed using Kruskal–Wallis H-test, followed by Mann–Whitney $U$-test with Bonferroni correction applied for the significance level ($P < 0.05$). Statistical analysis was performed in SPSS V.14 software for Windows (SPSS for Windows; SPSS Inc., Chicago, IL, USA).

## Taxonomic description

The characterization of *Pteroptyx* larvae follows *Ballantyne et al. (2011)*; *Ballantyne et al. (2015)*; *Ballantyne & Menayah (2002)* and *Jusoh et al. (2018)*, which primarily focused on body shape, tergite shape and pigmentation, spiracle position, antennae, mouthparts, and legs. In this study, additional characters were defined, such as dissected mouthparts, small

setae on the claw, prosternum pigmentation, and the number of retractable filaments in the holdfast organ. The larval characters of each species are based on the description for *P. valida* larvae by *Ballantyne & Menayah (2002)*. These larval characteristics were also compared with those of aquatic species (*Ballantyne et al., 2022*), which described the anterior and posterior ventral areas of the meso- and metathorax of larvae as basisternum and sternellum, respectively. Supplementary redescriptions of *P. valida* are included to facilitate clear comparisons among species.

## RESULTS

### Taxonomy

Lampyridae Rafinesque, 1815
Luciolinae Lacordaire, 1857
*Pteroptyx* E. Olivier, 1902
*Pteroptyx asymmetria* Ballantyne, 2001 (Figs. 1–2)
*Pteroptyx malaccae* (Gorham, 1880) (Figs. 3–4)
*Pteroptyx tener* E. Olivier, 1907 (Figs. 5–6)
*Pteroptyx valida* E. Olivier, 1909 (Figs. 7–8)

**Generic diagnosis.** Found in riparian locations within mangrove or blackish water areas near river mouths; terrestrial form; elongate, slender, spindle-shaped, without laterally explanate tergal margins, laterotergites visible at sides, bearing the spiracles are visible from above (*Ballantyne et al., 2011*; *Ballantyne et al., 2015*; *Fu, Ballantyne & Lambkin, 2012b*), posterolateral corners of terga 1–8 round or produced depending on species; dorsal surface smooth without tubercles or with tubercles depending on species; mandible without inner teeth; antennal segment 3 much shorter than the other antennal segments, sensory cone adjacent to segment 3 short, its size and shape varied; without brush of setae from apex of tibiotarsus; similar to all Luciolinae larvae such as *Colophotia* species (*Ballantyne et al., 2019*; *Ballantyne & Lambkin, 2009*, explained in Fig. 517) and *Curtos* (*Fu, Ballantyne & Lambkin, 2012a*), but with dark pigmentation of tergites. The first stage of larvae can be distinguished from the texture of tergites with uniformly coarse granular and brown color on the ventral side in *P. malaccae* and *P. tener*.

**Remarks.** As a member of the subfamily Luciolinae, a group with diverse firefly species, most members within the subfamily have distinct characteristics on the dorsal surface. However, many of them have a ventral surface similar to *Pteroptyx* (*Fu, Ballantyne & Lambkin, 2012a*). *Pteroptyx* has a slender shape similar to *Abscondita* and *Pygoluciola*, but have a wide pale median line on dorsal plates and lacks heavily sclerotized tergal plates (*Fu & Ballantyne, 2008*, explained in Fig. 2; *Nada, Ballantyne & Jusoh, 2021*, explained in Fig. 2.); shape similar to *Curtos*, but have pigmented tergal plates with pale median line (*Fu, Ballantyne & Lambkin, 2012a*, explained in fig. 36). *Pteroptyx* larvae differ from those of *Asymmetricata*, *Atyphella*, and *Emeia*, which have a protergum that is wider than it is long and laterally explanate tergal margins.

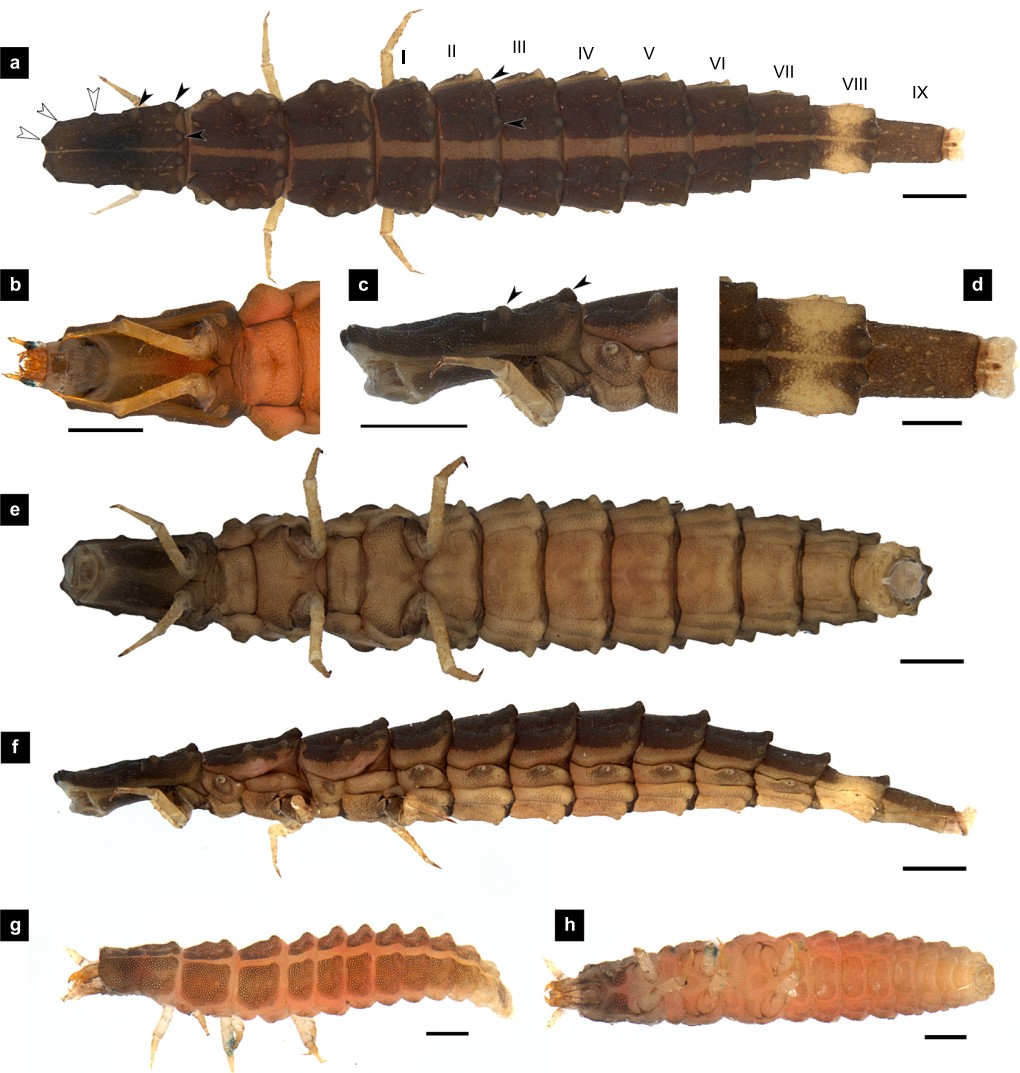

**Figure 1** **Morphological characters of *P. asymmetria* larvae.** (A–F) Fifth instar; (G–H) first instar. (A, D, G) Dorsal view; (B, E, H) ventral view; (C, F) lateral view. (B, C) head area; (D) tail area. White and black arrows denote small and large tubercle s, respectively. Roman numerals indicate abdominal segment numbers. Scale bar (A, C, E, F) 0.5 mm; (G–H) 0.01 mm.

**General characteristics of *Pteroptyx* last instar larvae.** (modified and expanded from *Ballantyne et al. (2011)*; *Ballantyne & Menayah (2002)*; *Fu, Ballantyne & Lambkin (2012b)* (explained in Figs. 55–60, 89–90); *Jusoh et al. (2018)*.

**Dorsal surface.** Three thoracic and nine obvious abdominal segments, narrow ring at posterior end of abdominal segments IX interpreted as segment X as holdfast organ (pygopodia) (*Ballantyne et al., 2022*; *Ballantyne & Menayah, 2002*; *Nada, Ballantyne & Jusoh, 2021*), elongate, fusiform, slightly flattened dorsoventrally, membranous and very soft bodied, lacking laterally explanate tergal plate; dorsal body plates with dark pigmentation, but not sclerotized, *P. valida* are well darker than the others; pale median

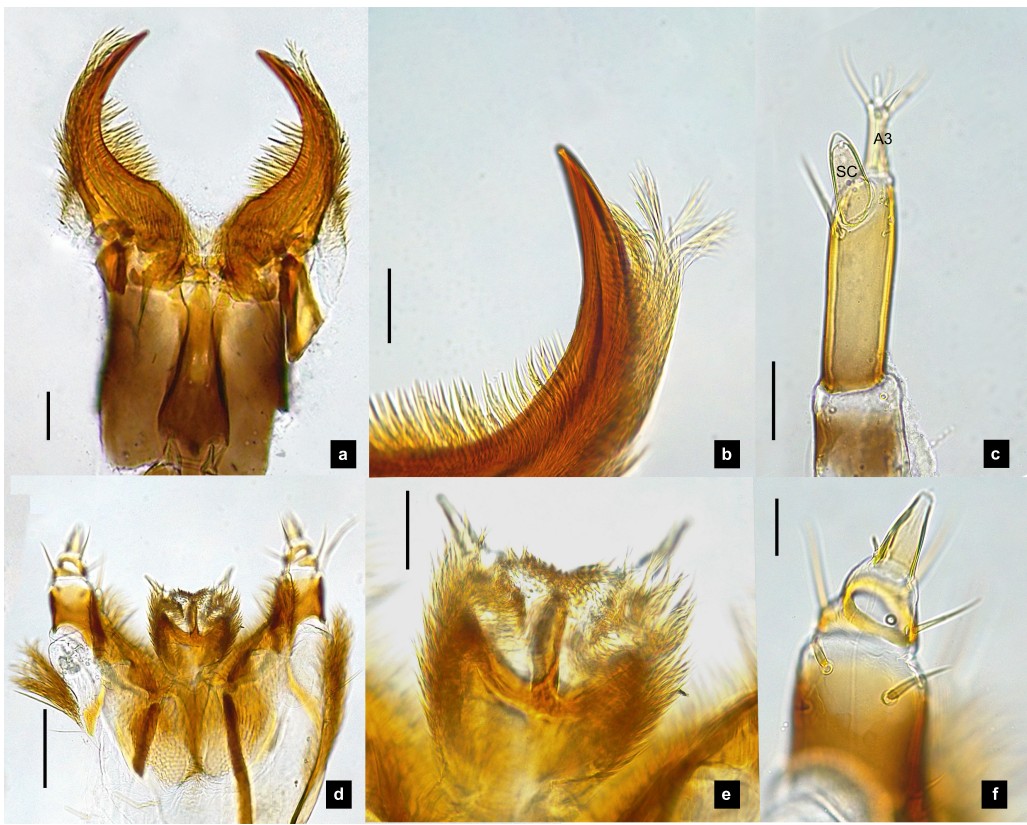

**Figure 2** **Mouthparts of *P. asymmetria* larvae, dorsal views.** (A–B) Mandibles; (C) antenna; (D) labium and maxilla complex; (E) labium; (F) maxillary palp. A3 and SC indicate antennal segment 3 and sensory cone, respectively. Scale bar: 0.05 mm.

line clearly visible from thorax to abdominal tergite IX, divided dorsal plates into two areas, each area emarginated on posterior margin; elevated along posterior tergal margins; texture of tergal plates smooth or bearing tubercles on lateral and posterior edge, varied in different species; all tergites brown except abdominal tergite VIII light tan or with brown pigmentation; epipleural plates on mesothorax and laterotergites on abdominal segments I–VIII bearing spiracles, all abdominal spiracles visible along the sides of body, except mesothoracic spiracles visible from above (Figs. 1A, 3A, 5A, 7A).

**Ventral surface.** An elongate pleural suture runs from anterior margin of mesothorax to posterior margin of abdominal segment IX; meso and metathorax subdivided into two areas, an anterior basisternum with paired laterotergites bearing spiracles on mesothorax and a posterior subrectanular sternellum bearing the legs (Fig. 7F) (*Ballantyne & Menayah, 2002*; *Fu, Ballantyne & Lambkin, 2012b*); abdominal segments subdivided into median sternal area with laterosternites at the sides, and paired laterotergites with spiracles above that just beneath the terga; pigmented sternellum area and laterosternites (*Ballantyne & Menayah, 2002*), found in *P. malaccae* and *P. valida* or absent in *P. asymmetria* and *P. tener*.

**Head.** (Figs. 3A–3C, 3F, 4A, 7A, 7B) Prognathous, subparallel sided, smooth; subrectangular, dorsoventrally flattened, well sclerotized; attached to an elongated

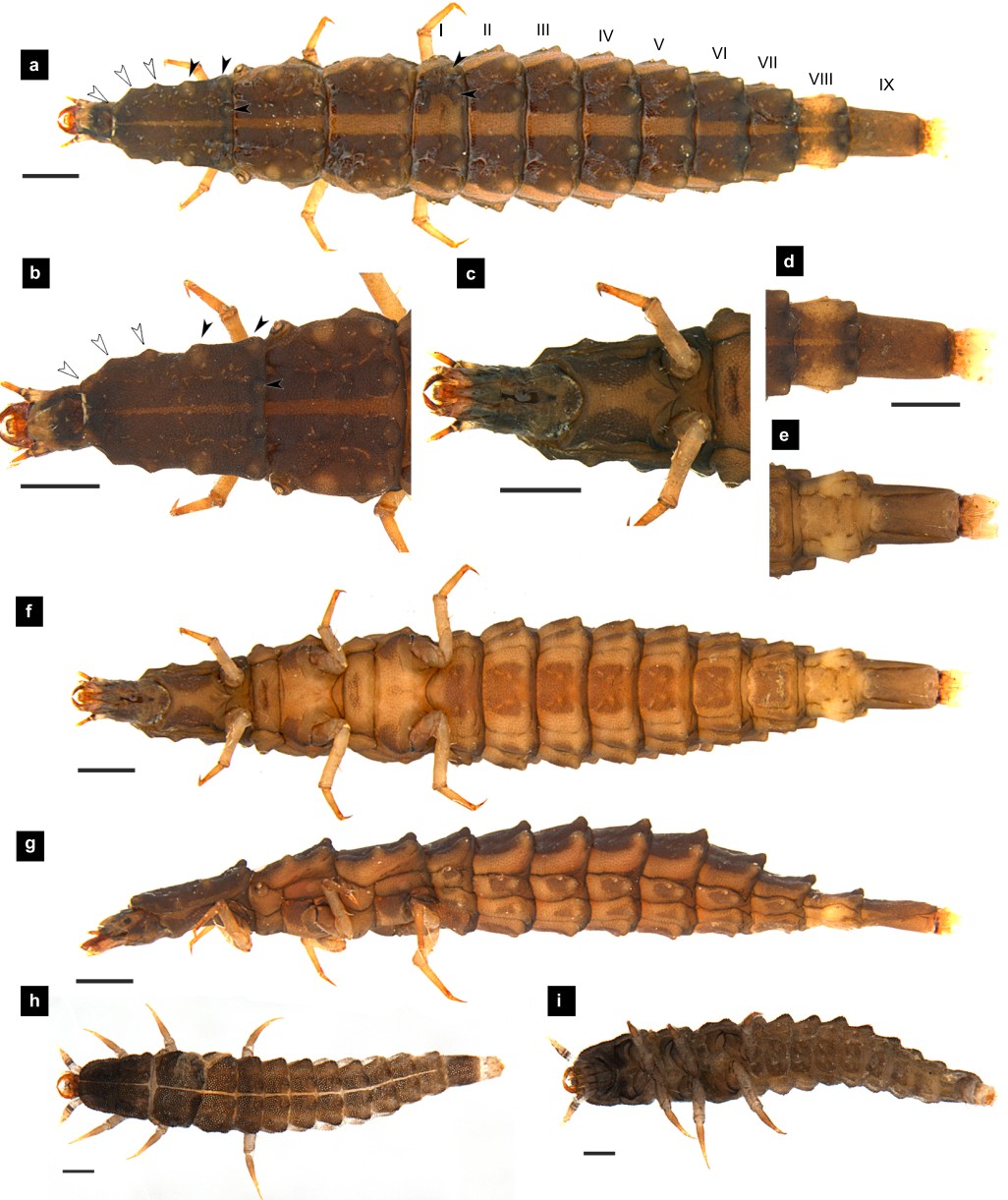

**Figure 3 Morphological characters of *P. malaccae* larvae.** (A–G) Fifth instar; (H–I) first instar. (A, B, D, H) dorsal view; (C, E, F, I) ventral view; (G) lateral view. (B, C) head area; (D, E) tail area. White and black arrows denote small and large tubercles, respectively. Roman numerals indicate abdominal segment numbers. Scale bar (A–G): 0.5 mm; (H–I): 0.01 mm.

extensible neck membrane, which allows the head to be extended out of or retracted within prothorax; with median dorsal frontoclypeus, bounded laterally and posteriorly by frontal arms of the ecdysial line; parietals reflexed ventrally; dark brown, antennal bases pale; maxillae and labium fused forming a maxillolabial complex covering most of ventral head area; a pair of lateral stemmata at the base of antennae.

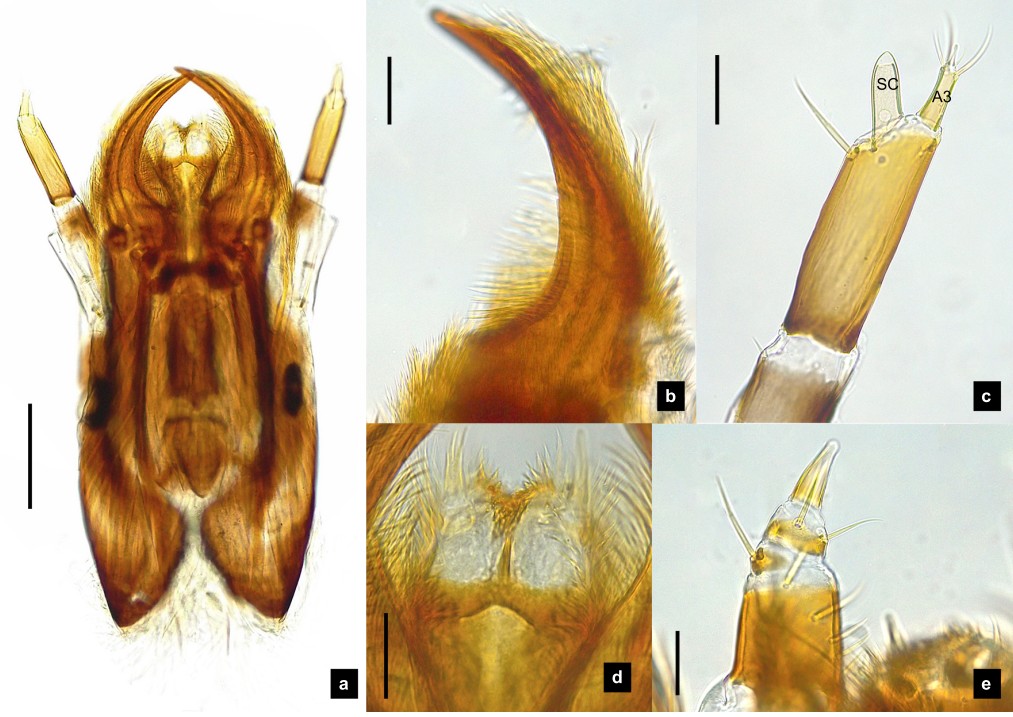

**Figure 4  Mouthparts of *P. malaccae* larvae, dorsal views.** (A) Head capsule; (B) mandible; (C) antenna; (D) labium; (E) maxillary palp. A3 and SC indicate antennal segment 3 and sensory cone, respectively. Scale bar. (A–E) 0.2 mm; (C–E) 0.05 mm.

**Antenna.** (Figs. 2C, 4A, 4C, 6C, 8B) Three-segmented, smooth and glabrous; originating on lateral-apical edges of head capsule adjacent to the mandibular bases (Fig. 4A); basal segment short and widest, attached to membranous base, smooth and glabrous, elongate articulating membrane creates a two-layered covering around the retracted antenna; second segment as wide as the basal one but much longer, elongated, sclerotized, smooth and glabrous, except a long subapical hair, apex obliquely truncate; third segment very short compared to the other segments (Figs. 2C, 4C, 6C, 8B), four short and one apical setae, a small elongate SC, SC subequal in length with A3 or little longer depends on species, on ventral side.

**Mandible.** (Figs. 2A–2B, 4A–4B, 6A–6B, 8A) Symmetrical, strongly falcate, tapering to a truncate apex with an inner channel opening subapically on outer edge (Figs. 2B, 8A); no toothed retinaculum; covered by long setae, especially outer side, long seta parallel in median region of ventral and dorsal inner side (Figs. 6A–6B), no setae at apical point of mandible near channel opening of canal. Maximum curvature of mandibles in middle of mandible.

**Labium.** (Figs. 2D, 4D, 6D, 8C–8D) Loosely attached to maxilla, short and strongly sclerotized prementum, covered by dense fine setae. Postmentum heart shaped, with each anterior corner bearing a palpus; palpus short, 2-segmented (including palpiger) and bearing a long single spine.
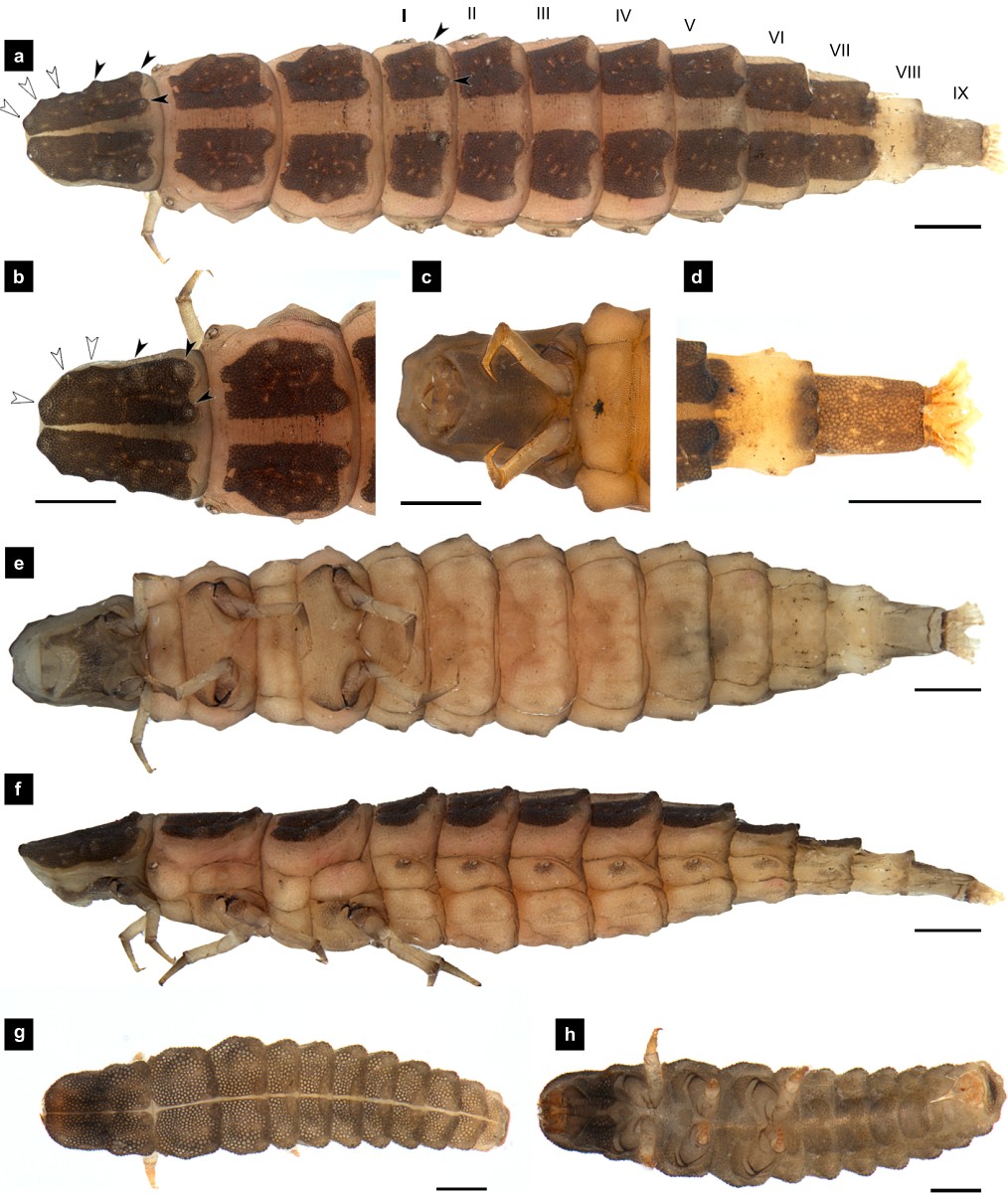

**Figure 5  Morphological characters of *P. tener* larvae.** (A–F) Sixth instar; (G–H) first instar. (A, B, D, G) dorsal view; (C, E, H) ventral view; (F) lateral view. (B, C) Head area; (D) tail area. White and black arrows denote small and large tubercles, respectively. Roman numerals indicate abdominal segment numbers. Scale bar (A–F) 0.5 mm; (G–H) 0.01 mm.

    **Maxilla.** (Figs. 2F, 4E, 6E, 8C, 8E) Long and robust, closely attached to labium. Stipes bearing dense long setae on outer side of maxilla. Galea large, 2-segmented, basal segment very long, lacking setae; distal segment short, narrowing distally and bearing a few short setae. Lacinia large, inner surface covered with setae. Palpus 4-segmented (including the palpifer), basal segment largest, subquadrate, longer than other segments, bearing a few

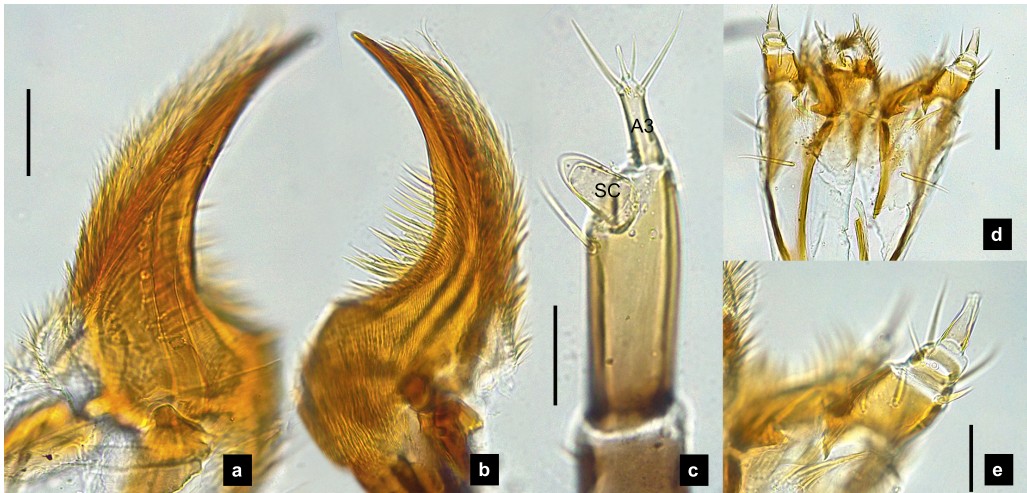

**Figure 6  Mouthparts of *P. tener* larvae.** (A) Dorsal mandible; (B) ventral mandible; (C) antenna; (D) labium and maxilla complex, dorsal view; (E) maxillary palp, dorsal view. A3 and SC indicate antennal segment 3 and sensory cone, respectively. Scale bar (A–C, E): 0.05 mm; (D): 0.1 mm.

long setae (Fig. 4E); distal segment cone-shaped, narrowing to a distal point, without setae, lacking a globular sensorium-type structure.

**Thorax.** (Figs. 1A–1C, 3A–3C, 3F–3G, 5A–5C, 5E–1F, 7A–7C, 7F–7G) Protergite subrectangular, longer than wide except *P. tener*, wider at the base (Fig. 5B), containing retracted head within it (Figs. 1B, 3G); meso- and metathorax subrectangular, shorter than prothorax; all thoracic tergites subdivided by median line, median line expands gradually from protergite to metatergite; each tergite elevated at posterior margin making slight slope upward (Fig. 1C). Texture of tergites differed across species, smooth without tubercles found only in *P. valida*; the other species have rough tergites with tubercles (Fig. S4). Color patterns of ventral surface differentiated across species (Fig. S4), no pigmented area in *P. asymmetria* and *P. tener* and with brown pigmented area in *P. malaccae* and *P. valida*; variation of pigmented areas appeared on prosternum across species; bearing well developed spiracles on mesothorax.

**Legs.** (Figs. 9A–9D) Four-segmented, coxae long and cylindrical; trochanters elongate; femur with a single short seta in middle, with the length of the seta varying among species; tibiotarsus as long as femora, covered with short, strong setae, with a single apical claw, tiny setae found in *P. malaccae*; prothoracic legs shorter than meso and metathoracic legs; legs pale with dark coxa.

**Abdomen.** (Figs. 1A, 1D–1F, 3A, 3D–3G, 5A, 5D–5F, 7A, 7D–7G) Segments wider than long, tapering towards end; each tergite subrectangular, tergites I–VIII divided by a median line, which divides tergites into two plates; posterior margin of each plate of tergite emarginate; surface of tergites may be smooth or rough with tubercles on posterior margin of tergite plates depend on species; pleural areas well developed, abdominal segments I–VIII subdivided; pleural areas of segments IX and X reduced; color pattern similar to that of thorax; brown abdominal tergites I–IX except VIII appearing different color

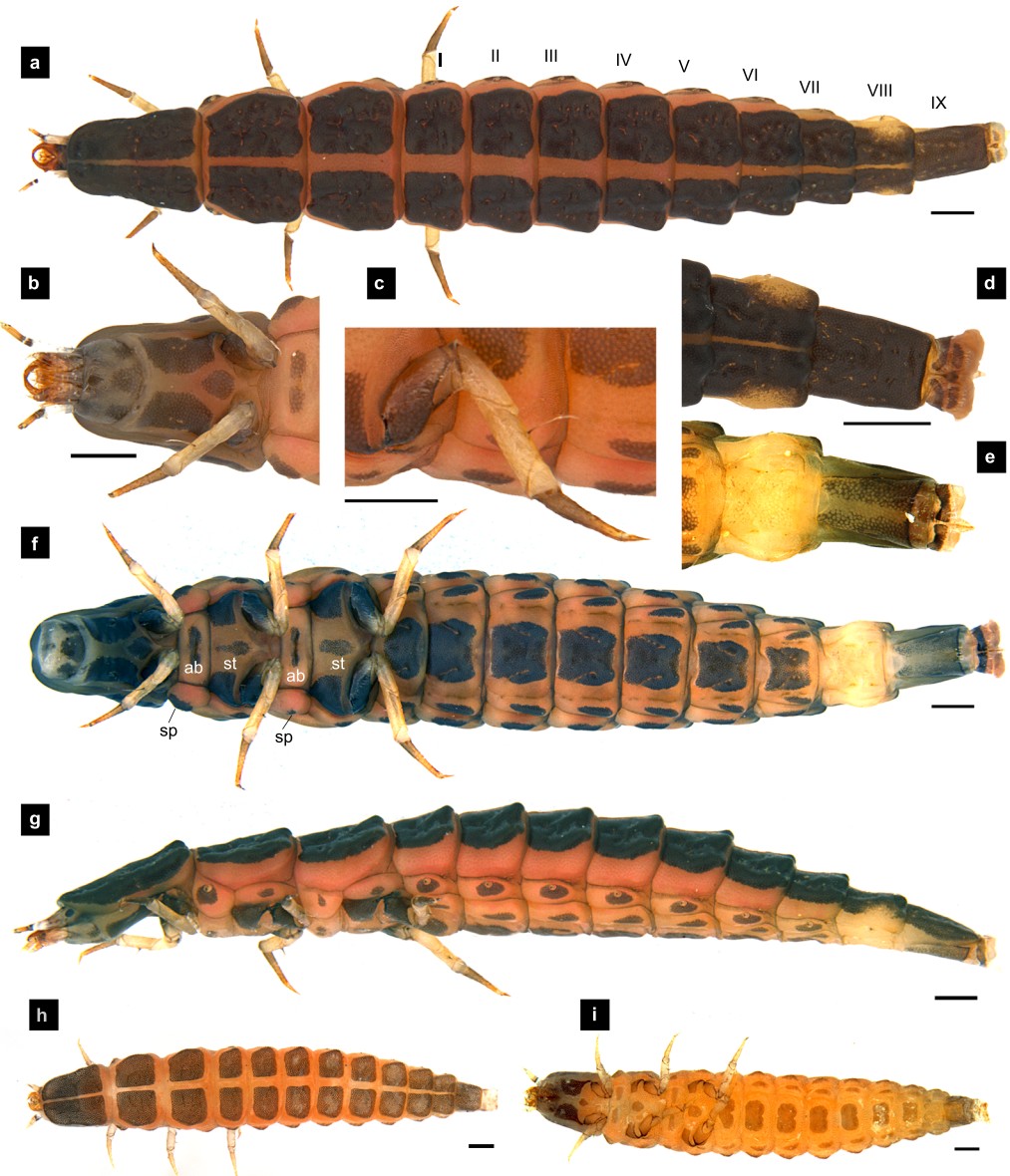

**Figure 7  Morphological characters of *P. valida* larvae.** (A–G) Sixth instar; (H–I) first instar. (A, D, H) Dorsal view; (B, C, E, F, I) ventral view; (G) lateral view. (B) Head area; (C) metathoracic leg; (D) tail area. ab, sp and st indicate anterior basisternum, spiracle and sternellum, respectively. Roman numerals indicate abdominal segment numbers. Scale bar (A–G) 0.5 mm; (H–I): 0.01 mm.

patterns in different species, bearing a pair of light organs on ventral side of segment VIII; segment IX elongate slender; segment X a narrow ring carrying holdfast organ, numbers of retractable filaments of holdfast organ differed in different species, *P. valida* has more than 10 filaments (Figs. 10A–10D), bearing different outline on the base of holdfast organ, a small spine underneath the holdfast filament (Fig. S3).

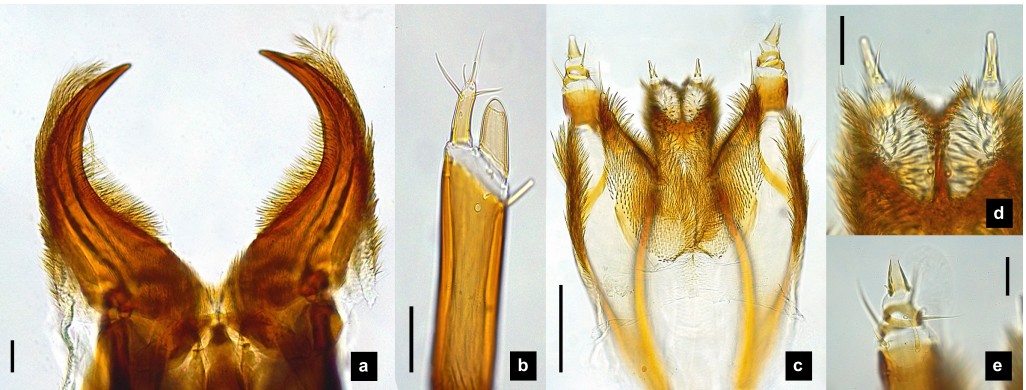

**Figure 8** **Mouthparts of *P. valida* larvae, dorsal views.** (A) Mandibles; (B) antenna; (C) labium and maxilla complex; (D) labial palps, (E) maxillary palp. A3 and SC indicate antennal segment 3 and sensory cone, respectively. Scale bar (A–D) 0.05 mm; (C–E) 0.2 mm.

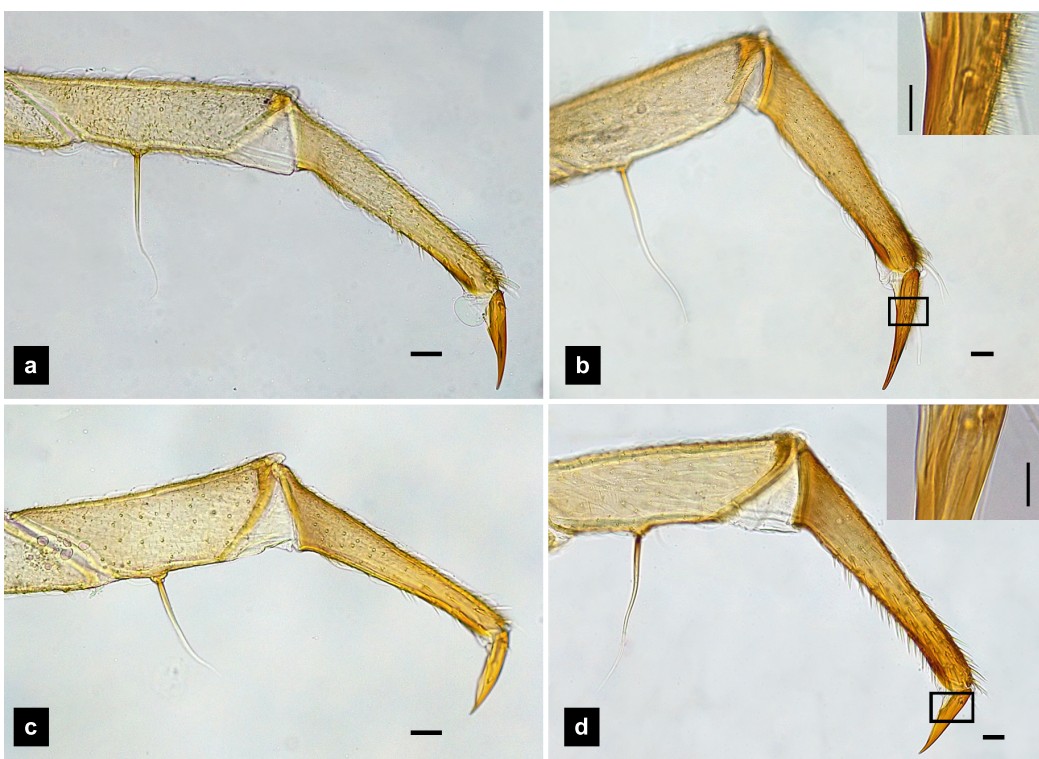

**Figure 9** **Metathoracic legs.** (A) *P. asymmetria*; (B) *P. malaccae*; (C) *P. valida*; (D) *P. tener*. The top right image is an enlargement of the claw area outlined in black. Scale bar: 0.25 mm.

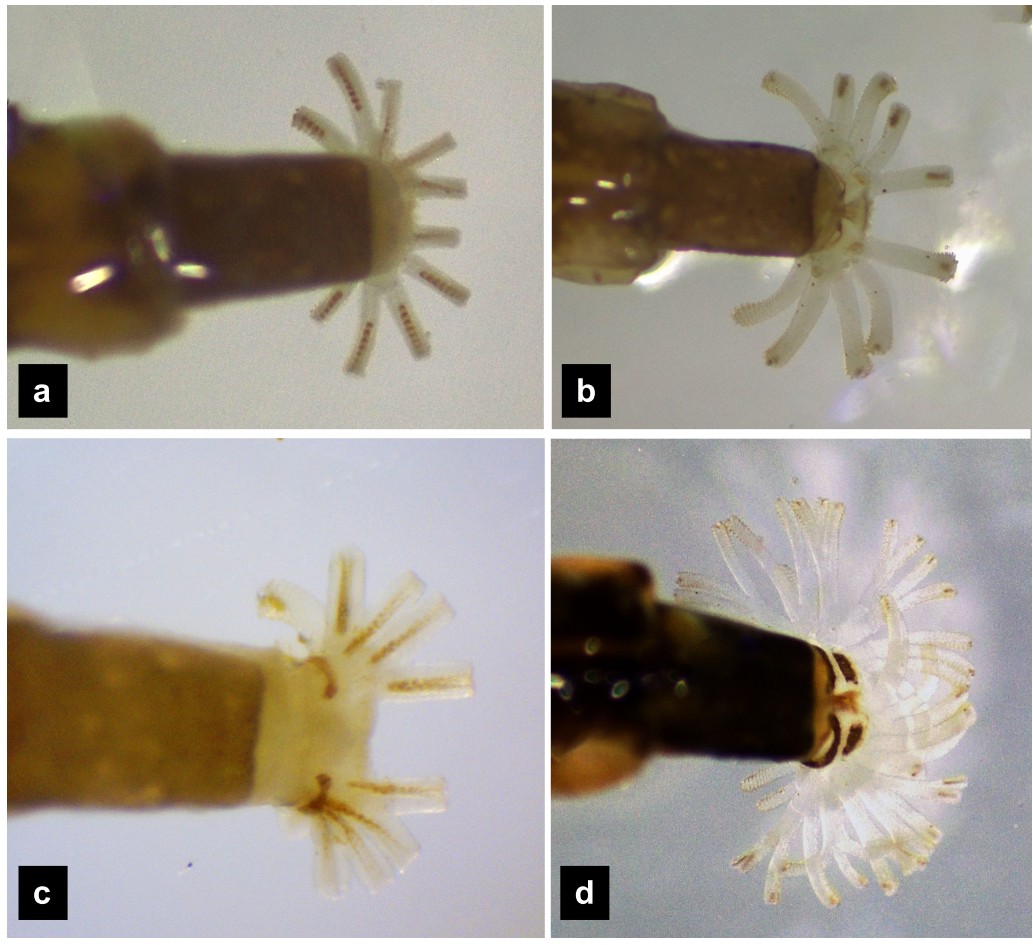

**Figure 10** **Holdfast organ, dorsal view.** (A) *P. asymmetria*; (B) *P. malaccae*; (C) *P. tener* and (D) *P. valida*.

***Pteroptyx asymmetria* Ballantyne, 2001**

**Diagnosis.** *Pteroptyx asymmetria* can be distinguished from other *Pteroptyx* species, except *P. tener* by its smaller size; differing from *P. valida* by bearing tubercles on tergite 1–11 margin, with higher tubercles than those of *P. maipo*, having much longer protergum length than width; anterior margin of protergum angles not round as in *P. valida* and *P. maipo*; ventral surface dull color with no dark pigmentation; prosternum darkly pigmented except longitudinal median region (Figs. 1B, 1E), similar to *P. malaccae* and some *P. tener* (Type I, Fig. S4); abdominal tergite VIII light tan on lateral margin where located a pair of light organs, very similar to *P. malaccae*; holdfast organ bearing 10 retractable filaments, differing from *P. valida*.

**Description, fifth instar larvae.** (Figs. 1–2) 6.71–8.41 mm long; 1.31–1.69 mm wide; long protergum length than width (0.77–1.33 mm long; 0.73–0.99 mm wide; length:width 1.05–1.48); bearing tubercles on tergite 1–11 margin (Figs. 1A, 1C), six small tubercles

(white arrows) on anterior margin and six large tubercles (black arrows) on posterior margin of protergites, six large tubercles on lateral and posterior margin of meso- and metatergites, four big tubercles on posterior margin of abdominal tergites; tubercles well developed around protergite plates making anterior shaped angles of protergum, anterior margin of protergum shaped angles; dark marked prosternum except longitudinal median region, sometimes (20%) found a pigmented diamond shape in posterior region; femora have one single short setae in middle, 0.26 mm long, longer than a half of tibiotarsus (Fig. 9A); abdominal tergites I–IX brown, except tergite VIII light tan on lateral margin where located a pair of light organs; no color pigmentation on ventral surface except slightly pigmented on coxa; holdfast organ bearing 10 retractable filaments, with two curved outlines on the base sides; SC subequal in length with A3 (Fig. 2C).

**Description, first instar larvae.** (Figs. 1G, 1H) 0.95–1.08 mm long; all dorsal plates uniformly coarse granular except tubercle areas, tubercles on all tergites formed except small tubercles on protergite, each tergites elevated at the posterior margin; median line on thoracic segments and abdominal tergites I–VII; ventral surface light brown except brown pigmentation on prosternum; no color pigmentation on ventral surface of meso- and metathorax; uniformly fine granular distributed on ventral area.

### *Pteroptyx malaccae*  (Gorham, 1880)

**Diagnosis.** One of three species of *Pteroptyx* known from Thailand and *P. maipo* with tubercles on tergite 1–11 margin, differing from both *P. valida*; protergum longer than wide, considerably similar to *P. tener* and *P. maipo*, but with protergum length not as long as *P. asymmetria*; having protergum shaped angles on anterior margin and distinguished from *P. valida* and *P. maipo* which have rounded shaped angles; one of three species possessing brown pigmented areas on the ventral side, similar to *P. maipo* and *P. valida*, differing from *P. valida* by darker pigmentation; prosternum dark pigmented except longitudinal median region, distinguished from some *P. asymmetria* (20%) and *P. tener* (Type I) by bearing a pigmented diamond shape in posterior region; claw bearing very small setae that differed from the other species; abdominal tergite 8 with brown; holdfast organ bearing 10 retractable filaments, differing from *P. valida*.

**Description, fifth instar larvae.** (Figs. 3–4) 8.27–10.97 mm long; 1.78–2.03 mm wide; protergum length longer than width (1.25–1.58 mm long; 1.05–1.23 mm wide; length:width 1.03–1.23); short tubercles on tergite 1–11 margins (Figs. 3A–3B), twelve tubercles around protergite, six tubercles on lateral and posterior margins of meso- and metatergites, coxa brown pigmented on ventral surface; femora bearing one single short setae in middle, 0.43 mm long, longer than three-quarters of tibiotarsus (Fig. 9B), claw bearing minute setae; four tubercles on posterior margin of abdominal tergites; abdominal tergite VIII pale with brown pigmentation (Figs. 3A, 3D); abdominal tergites I–IX brown except tergite VIII tan colored on lateral margins adjacent to the paired larval light organs, also visible in ventral view; holdfast organ bearing **10** retractable filaments; SC = 2/3A3 (Fig. 4C).

**Description, first instar larvae.** (Figs. 3H–3I) 2.25 mm long; all dorsal plates uniformly coarse granular except tubercle areas, tubercles on all tergites formed incompletely; median line on thoracic segments and abdominal tergites I–VII; ventral side with dark pigmentation on prosternum; uniformly fine granular distributed on ventral area.

*Pteroptyx tener* **(Olivier, 1907)**

**Diagnosis.** One of three *Pteroptyx* with tubercles on the margin of tergites 1–11; distinguished from *P. asymmetria* and *P. malaccae* by subequal protergite; a light tan ventral surface distinguishes this species from *P. maipo*, *P. malaccae* and *P. valida* by lacking brown pigmented areas as well as a brown pigmented area in the medial region of abdominal tergite VIII; holdfast organ bearing ten retractable filaments.

**Description, sixth instar larvae.** (Figs. 5–6) 5.78–8.41 mm long; 1.24–1.82 mm wide; length:width 0.99–1.20); small size (7.35–8.34 mm long; 1.52–1.84 mm wide); bearing tubercles on margins of tergites 1–11 (Fig. 5A–5B), six small tubercles on anterior margin and six large tubercles on posterior margin of protergite, six large tubercles on lateral and posterior margin of meso- and metatergites, four large tubercles on posterior margin of abdominal tergites; anterior margin of protergum shaped angles; pale abdominal sternites lacking dark pigmentation (Fig. 5E). Ventral color pattern of prosternum variable: Type I (Fig. 5C, Fig. S4) prosternum darkly pigmented except longitudinal median region (four of six larvae), Type II (Fig. 5E, Fig. S4) prosternum ventral surface uniformly brown except a deeply emarginate pale area region at the anterior margin (two of six larvae); ventral surface from mesothorax rearward, without pigmentation; femora have one single short setae in middle, 0.20 mm long, equal half of tibia (Fig. 9C); abdominal tergites I–VII and IX brown, tergite VIII light tan in color, with pigmentation restricted to only anterior and posterior margins (Fig. 5D); ventral surface without pigmentation; holdfast organ bearing ten retractable filaments (Fig. 10C); SC = 3/4 A3 (Fig. 6B).

**Description first instar larvae.** (Figs. 5G–5H). 0.95–1.88 mm long; dorsal and ventral brown; all dorsal sclerites uniformly coarse granular, tubercles absent; median line on thoracic segments and abdominal tergites I−VII; ventral side very pale except brown prosternum; ventral surface uniformly finely granular.

*Pteroptyx valida* **E. Olivier 1909**

The larval characters of the first to third, as well as last instar larvae were examined and comprehensively described following the previous work of *Ballantyne & Menayah (2002)*. A supplementary redescription is provided here for comparison to the other *Pteroptyx* species known from Thailand in the above sections.

**Diagnosis.** One of two *Pteroptyx* species that have smooth tergites, lacking tubercles on all tergites, thereby differing from *P. asymmetria*, *P. malaccae* and *P. tener*; very similar in

dorsal and ventral shape and coloration to *P. maipo* but differing in having a pigmented diamond pattern on the posterior region of the prosternum, pale abdominal tergites VIII and pale lateral margins of abdominal tergite IX; ventral surface darkly pigmented; holdfast organ bearing more than 10 retractable filaments and with dark outline on the base sides.

**Redescription, sixth instar larvae.** (Figs. 7–8) 9.67–12.28 mm long; 1.83–2.54 mm wide; protergum length greater than width (1.2–1.80 mm long; 1.13–1.53 mm wide; length:width 1.07–1.18); tergites smooth, thoracic and abdominal sternites darkly pigmented, prosternum bearing a small pigmented diamond shape in the medial posterior region (Figs. 7B, 7F); anterior margin of protergum bluntly rounded (Fig. 7A); bearing dark pigmentation on ventral side on coxa and medial abdominal sternite; femora have one single short setae in middle, 0.39 mm long, longer than a half of tibia (Fig. 9D); dark brown abdominal tergites I–IX except tergite VIII which is light tan in the lateral regions at location of larval light organs (Fig. 7D); holdfast organ bearing more than 10 retractable filaments (Fig. 10D, Fig. S3A) and with dark outline at the base (Figs. 7D, 7F, 10D), with four curved outlines on the base sides; SC subequal in length with A3 (Fig. 8B).

**Redescription, first instar larvae.** (Figs. 7H–7I). 1.48–2.54 mm long; all dorsal sclerites uniformly coarse granular, tergites lacking tubercles; median line on thoracic segments and abdominal tergites I–VII; bearing dark pigmentation on ventral surface of prosternum, same as the sixth instar.

## A key to species of *Pteroptyx* last instar larvae known to occur in Thailand

1. Tergites smooth without tubercles; protergum length much longer than wide (Figs. 7A–7G) ……………………………………………………………*Pteroptyx valida* Olivier 1909

Tergites rough with tubercles; protergum length longer than wide or subequal (Figs. 1A, 3A, 5A)……………..………………………………………………………..…2

2. Protergum length longer than wide (long:wide > 1.25);abdominal tergite VIII lightly pigmented toward the lateral margins, dorsal medial area light brown (Figs. 1A–3F) ……………………………………..…….…*Pteroptyx asymmetria* Ballantyne, 2001

Protergum width subequal in length (long:wide < 1.25); abdominal tergite VIII lightly pigmented toward the lateral margins or entirely lacking pigmentation (Figs. 3B, 5B)…..……3

3. Ventral surface lacking dark pigmentation; abdominal tergite VIII lacking pigmentation (Figs. 5A–5F) …………………….....………………*Pteroptyx tener* (Olivier, 1907)

Ventral surface with dark pigmentation; abdominal tergite VIII lightly pigmented on lateral margins, and light brown in medial area between the lighter pigmented margins (Figs. 3A–3G) ………………......................................................... *Pteroptyx malaccae* (Gorham, 1880)

## Morphological comparison among *Pteroptyx* species

Several differences in the morphological characters of *Pteroptyx* larvae were investigated (Table 2, Fig. S4). The characters determined to be important for species identification are protergum shape, tergite texture and coloration. Variation in prosternum coloration was found in the rough tergite species. The scatter plot (Fig. 11A) of protergum length and width contrast with the variation in size among *Pteroptyx* species. *Pteroptyx valida* larvae

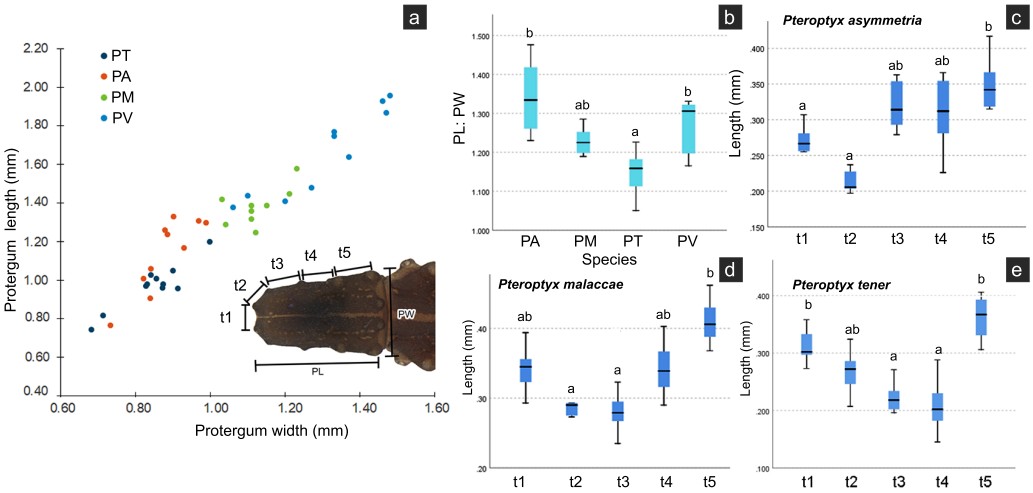

**Figure 11** **Comparison of morphological characters of *Pteroptyx* larvae.** (A) Scatter plot of protergum width (PW) and protergum length (PL), (B) PL:PW ratio, (C)–(E) distances between tubercles t1 to t5. PT, PA, PM, PV, PL, PW, PL:PW refer to *P. tener*, *P. asymmetria*, *P. malaccae* and *P. valida*, respectively. t indicates the distance between tubercles (T) in each position on the protergum. a, b indicates significant differences (Fisher's LSD test for multiple comparison tests) in tubercle distances at each position on protergum (*P* < 0.05).

have the largest protergum, followed *P. malaccae*, *P. asymmetria* and *P. tener*. *Pteroptyx asymmetria* exhibits the largest protergum length and largest PL:PW ratio, while *P. tener* is the smallest (Fig. 11B).

As three of the four species bear tubercles on all margins of the protergum, *P. asymmetria*, *P. malaccae*, and *P. tener* were analyzed for the distances between the tubercles in each species. T1 to T5 in each species were significantly different (Figs. 11C–11E; Kruskal–Wallis H-test: $\chi 2 = 26.120$, $df = 4$, $P = 0.000$ for *P. asymmetria*; $\chi 2 = 24.537$, $df = 4$, $P = 0.000$ for *P. malaccae*; $\chi 2 = 31.754$, $df = 4$, $P = 0.000$ for *P. tener*). The distances of t1, t2, t3, t4 and t5 of *P. asymmetria* and *P. malaccae* follow a similar trend with t2 being the shortest and t5 being the longest, in contrast to *P. tener*.

## DISCUSSION

The taxonomic study of *Pteroptyx* firefly species based on both morphological and molecular analyzes has been underway for some time (*Ballantyne et al. 2019*; (*Ballantyne et al., 2015*; *Ballantyne & McLean, 1970*; *Jusoh et al., 2014*; *Jusoh et al., 2018*); however, with the exception of *P. valida* (*Ballantyne & Menayah, 2002*; *Ohba & Sim, 1994*) and *P. maipo* (*Ballantyne et al., 2011*) only adults were considered. Both of these species were placed into the *Pteroptyx* "terrestrial group" due to absence of abdominal gills and terminal spiracles in their larvae (*Ballantyne et al., 2011*; *Fu, Ballantyne & Lambkin, 2012b*).

The shared characteristics of the previously described *Pteroptyx* larvae, *P. valida* and *P. maipo*, include being slender, lacking laterally explanate tergal margins, exhibiting visible laterotergites from above, featuring a narrow median longitudinal line, lacking teeth on the mandibles, and possessing three antennal segments with an elongated sensory

Boonloi et al. (2025), *PeerJ*, DOI 10.7717/peerj.19190

**Table 2  Comparison of last instar larvae of *Pteroptyx*.**

| Character | *Pteroptyx* species | | | |
|---|---|---|---|---|
| | *P. asymmetria* | *P. malaccae* | *P. tener* | *P. valida* |
| Pronotal shape | PL > PW | PL ≥ PW | PL = PW | PL > PW |
| Ratio of PL:PW | 1.31 ± 0.14 | 1.22 ± 0.08 | 1.10 ± 0.07 | 1.12 ± 0.04 |
| Ratio of SC: A3 | 1.12 | 1.58 | 1.37 | 1.00 |
| Body length (mm) | 6.71–8.41 | 8.38–9.54 | 5.78–8.41 | 9.67–12.28 |
| Tergite texture | With tubercles | With tubercles | With tubercles | Smooth, without tubercles |
| Pronotal anterior margin | Angled | Angled | Angled | Rounded |
| Prosternum pigmentation | Darkly pigmented except median region, with diamond shape in posterior region (80%) | Darkly pigmented except median region, with diamond shape in posterior region (90%) | Darkly pigmented except median region (66%) or crescent-shape (34%) | Darkly pigmented except median with posterior diamond shape (100%) |
| Ventral color pattern | Brown without pigmented areas | Light brown with pigmented areas | Light brown without pigmented areas | Light brown with pigmented areas |
| Tergite 11 | No pigmentation on lateral margins | No pigmentation on lateral margins | No pigmentation across almost the entire segment | No pigmentation on lateral margins, but area is smaller than PA and PM |
| Seta on metathoracic femur | 0.26 mm long, longer than a half of tibia | 0.43 mm long, longer than three-quarters of tibia | 0.20 mm long, equal half of tibia | 0.39 mm long, longer than a half of tibia |
| Tibiotarsus | Smooth, no setae | Very tiny setae | Smooth, no setae | Smooth, no setae |
| Holdfast organ | 10 retractable filaments | 10 retractable filaments | 10 retractable filaments | More than 10 retractable filaments |

cone on antennal segment 3 (*Ballantyne et al., 2015*). Our study of the larval morphology of additional *Pteroptyx* species expands upon this previous work and contributes new characters useful for identifying *Pteroptyx* larvae to the species level.

Larvae of the family Lampyridae are known to bear spiracles on the mesothoracic and abdominal segments I–VIII (*Ballantyne et al., 2011*; *Branham & Archangelsky, 2000*; *Zurita-García et al., 2022*). However, in *Pteroptyx* larvae, the spiracles on the mesothorax (*P. asymmetria*, Fig. 1; *P. malaccae*, Figs. 3A–3B) appear larger and are located more dorsally compared to spiracles on the abdomen, or the mesothoracic spiracles of other lampyrid genera (*Branham & Archangelsky, 2000*; *Fu, Ballantyne & Lambkin, 2012b*; *Vaz et al., 2021*; *Zurita-García et al., 2022*). While this feature of the mesothoracic spiracles may appear to be a prominent characteristic of *Pteroptyx* species, they are positioned more laterally in *P. valida* and *P. maipo*, a shared similarity with the known larvae of other firefly genera. Further studies are needed to investigate whether this seemingly modified condition of the mesothoracic spiracles might have altered respiratory functions.

Our study yielded additional variation in larval morphology both across and within these species. Both tergite texture and the color patterns of the ventral area, prosternum, and tergite VIII appear to represent species-specific characters. However, intraspecific variation in the color pattern of the prosternum was observed in *P. asymmetria*, *P. malaccae*, and *P. tener* within the same populations of each species. Therefore, this character alone is not adequate to precisely identify a larva to species. It should be noted that morphological variation in *P. malaccae* adults was also found in different populations or regions (*Jusoh et al., 2018*); therefore, it seems reasonable that the variation observed in these larval characters may also be present in different populations or regions. Other larval color patterns are much less variable. The differences in color patterns of abdominal tergite VIII and IX between *P. maipo* and *P. valida* were investigated by *Ballantyne et al. (2011)*, and appear both appropriate and useful for distinguishing larvae in this genus at the species level.

The size of larval characteristics can be useful characters for species identification of *Pteroptyx* species. The last instar larvae of *P. valida* are the largest, followed by *P. malaccae*, while *P. asymmetria* and *P. tener* fall within a similar size range. *Pteroptyx maipo* with a length of 13 mm (*Ballantyne et al., 2011*), falls within the large size class, similar to *P. valida*. However, identifying field-collected specimens to the species level can be challenging due to unknown growth stages (larval instars). We have only treated first and last instar larvae in this paper. The protergum shape, specifically the comparison of PL and PW, can aid in identifying *P. tener*, as it exhibits only subequal PL and PW, whereas the other species have PL longer than PW. In this study, variations in the morphometrics of these species were observed under magnification. However, it needs to be realized that the chilling process prior to measurement might yield information that differs from that of live larvae in the field without any treatment.

This variation included the size of the sensory cone on antennae, claw texture, and the number of retractable filaments on the holdfast organ, which have not yet been reported.

*Ho (2002)* investigated the variation in and the arrangement of filaments of the holdfast organ among different species, which was later confirmed by *Fu, Ballantyne*

*& Lambkin (2012b)*. This structure has not been described from *Pteroptyx* species, until now. Differences in the number of retractable filaments on the holdfast organ were found across various *Pteroptyx* species, with *P. valida* displaying a significantly higher number of filaments (>10) compared to other species (=10). The number of filaments may be related to larval habitats or behavior, but remains unknown. Our findings are consistent with the scanning electron microscope (SEM) photos of *Fu, Ballantyne & Lambkin (2012b)*, which showed that terrestrial larvae bear more filaments than those with a semiaquatic or aquatic lifestyles. Additionally, *Zurita-García et al. (2022)* found 30 filaments from 12 basal stalks of the holdfast organ in another terrestrial species, *Photinus extensus*. The holdfast organ serves multiple functions in the larval stage (*Archangelsky & Branham, 1998*), including adhering to substrates during movement (*Fu et al., 2005*), supporting predatory behavior (*Sato & Yoshikawa, 2024*), cleaning mucus of slugs and snails (*Cyril & Joseph, 2023*), and building the pupal cell. These observations suggest that *P. valida* may have a different life history compared to other *Pteroptyx* species, possibly indicating a more terrestrial lifestyle. It is intriguing to consider this character for larval identification and taxonomy. However, SEM techniques are expensive and may not adequately display outspreading filaments. Therefore, this study suggests photographing live larvae while they are submerged under water, but this could be challenging in species with sensitive behavior like *P. asymmetria*.

The morphological diversity of larval mouthparts across taxa are hypothesized to be related to species feeding ecology, as was shown in carabid beetle larvae (*Brandmayr et al., 1998*). We assume this is also true for *Pteroptyx* species. The mandibles of firefly larvae have long been understood to have an inner channel with an opening near the apical area (*Branham & Archangelsky, 2000*) that is used for injecting midgut secretions which both paralyze prey as well as facilitating extra-oral digestion of prey tissue (*Branham & Archangelsky, 2000*; *Krämer, Hölker & Predel, 2024*). The morphology of larval mandibles within this firefly genus, appear modified for prey capture, *e.g.*, variation in curvature and the presence of the retinaculum across various species. The retinaculum was mostly present in semiaquatic (*Pygoluciola*) (*Nada, Ballantyne & Jusoh, 2021*; *Fu, Ballantyne & Lambkin, 2012b*) and terrestrial larval species such as *Abscondita* (*Ballantyne et al., 2019*), *Lucidota* (*Branham & Archangelsky, 2000*), *Photuris* (*Rosa, 2007*), *Psilocladus* (*Vaz, da Silveira & Rosa, 2020*), and *Pyropyga* (*Archangelsky & Branham, 2001*), while it appears absent in aquatic species such as *Aquatica* and *Sclerotia* (*Fu, Ballantyne & Lambkin, 2012b*). The retinaculum is also absent in *Pteroptyx*, similar to *Aquatica* and *Sclerotia*.

Variation in the antennae of beetle larvae have been identified in other families of Coleoptera. However, in *Pteroptyx* the morphology of the third antennal segment, which has setae at the apex, is similar to those in terrestrial and semiaquatic groups in the Lampyridae, according to *Fu, Ballantyne & Lambkin (2012b)*. The sensory cone of the antennae might be important for the perception of olfactory stimuli in the air, as seen in various kinds of terrestrial insect larvae (*Akent'eva, 2011*), but this does not appear necessary for predation under water. A careful examination of the sensory cone, sensilla and surface ultrastructure could potentially contribute to a better understand the feeding habits of *Pteroptyx* firefly larvae.

Our findings have documented general characteristics of the known larvae of *Pteroptyx* species, including likely species-specific characteristics that could serve as easily observable diagnostic features useful for identification to the species level. Additionally, we have provided photographic plates (Fig. S4) as references to document characters and character states and an identification key for distinguishing the larvae of these four species.

It is our hope that these larval characters might serve as the foundation for developing an expanded dataset useful for delimiting *Pteroptyx* species, as well as documenting larval morphology and its variation. The ability to accurately identify firefly larvae can be applied to studies of behavior, biodiversity and surveying areas of conservation concern through the assessment of both adult and larvae. In addition, this knowledge could be applied to the challenges of conducting surveys for adults congregated on very tall trees, where traditional collecting methods such as sweep-netting are impractical.

Knowledge of habitat requirements is crucial for firefly conservation as it is necessary for effective habitat management, as demonstrated in *Nipponoluciola cruciata* (Takeda et al., 2006). Each species likely has specific requirements to reduce niche overlap. Therefore, species-specific information on microhabitat requirements is essential for the conservation of larval habitats and the success of habitat restoration programs for *Pteroptyx* fireflies. For example, with rare species like *P. asymmetria*, there is insufficient data for IUCN Red List assessment. These species have very small populations that share habitats with other *Pteroptyx* species. Integrating survey information on the habitats of both adults and larvae is necessary to provide an accurate understanding of the species' true status. Specific habitat management could play a key role in conserving these firefly populations and reducing the risk of their extinction.

As a group of highly charismatic insects, *Pteroptyx* fireflies play an important role in raising awareness and advocating for the conservation of mangrove forests. Furthermore, as an umbrella species, conserving their habitats also helps protect numerous other species that depend on these ecosystems. Identifying firefly species at the larval stage is crucial for understanding their microhabitats, which is essential for conservation of breeding habitats.

## CONCLUSIONS

The larval descriptions for three *Pteroptyx* species—*P. asymmetria*, *P. malaccae*, and *P. tener*—and the redescription of *P. valida* larvae were produced to enhance our understanding of larval stages within the lampyrid genus *Pteroptyx*. Readily observable species-specific characteristics were identified, such as size, texture, and color patterns of tergites and sternites, for use in the species identification of larvae. These characters are herein documented and incorporated into an identification key to larvae. Furthermore, measurements including PL:PW ratio, SC:A3 ratio, claw characteristics and the number of retractable filaments of the holdfast organ were documented and compared to support larval identification. *Pteroptyx valida* can be easily distinguished by its size, smooth tergites, and dark ventral surface. In contrast, *P. tener* lacks a pigmentation on the ventral surface and has a protergum of nearly equal size. *P. malaccae* and *P. asymmetria* exhibit similar morphology, differing primarily in body size and ventral color pattern.

**Abbreviations for taxonomic characters**

| | |
|---|---|
| **A3** | the third antennal segment |
| **ab** | anterior basisternum |
| **BL** | body length |
| **BW** | body width |
| **KU** | Department of Entomology, Faculty of Agriculture, Kasetsart University |
| **PA** | *P. asymmetria* |
| **PL** | protergum length |
| **PL: PW** | the ratio of protergum length to protergum width |
| **PM** | *P. malaccae* |
| **PT** | *P. tener* |
| **PV** | *P. valida* |
| **PW** | protergum width |
| **SC** | sensory cone of antenna |
| **sp** | spiracle |
| **st** | sternellum |
| **T1, 2,** *etc* | tubercle on protergum |
| **t1, 2,** *etc* | distance between tubercles |
| **I, II, III,** *etc* | abdominal segment number |

# ACKNOWLEDGEMENTS

The authors would like to thank Miss Rinrada Jundasri, Miss Phakaphon Chongchit, Miss Benyaphon Jarrukonchayapon, and Mr. Fapratan Sompomtip for collecting and raising the firefly larvae used in this study. We also thank Miss Natthareeya Boonyueng for her work on the preliminary study. We also extend our gratitude to Mr. Saichon Sohsiw for introducing us to the wonderful study site of *P. tener* and for his hospitality.

## Funding

This study was financed by the Kasetsart University Research and Development Institute (KURDI) (Grant No. FF(KU) 51.68) and Electricity Generating Authority of Thailand (EGAT). The funders had no role in study design, data collection and analysis, decision to publish, or preparation of the manuscript.

## Grant Disclosures

The following grant information was disclosed by the authors:
The Kasetsart University Research and Development Institute (KURDI): FF(KU) 51.68.
Electricity Generating Authority of Thailand (EGAT).

## Competing Interests

The authors declare there are no competing interests.

## Author Contributions

- Suparada Boonloi performed the experiments, analyzed the data, prepared figures and/or tables, and approved the final draft.
- Parichart Laksanawimol conceived and designed the experiments, performed the experiments, analyzed the data, prepared figures and/or tables, authored or reviewed drafts of the article, and approved the final draft.
- Soraya Jaikla performed the experiments, analyzed the data, prepared figures and/or tables, authored or reviewed drafts of the article, and approved the final draft.
- Marc A Branham analyzed the data, authored or reviewed drafts of the article, and approved the final draft.
- Anchana Thancharoen conceived and designed the experiments, performed the experiments, analyzed the data, prepared figures and/or tables, authored or reviewed drafts of the article, materials and laboratory, and approved the final draft.

## Ethics

The following information was supplied relating to ethical approvals (i.e., approving body and any reference numbers):

The research was approved for animal care and use for scientific research at Kasetsart University (ACKU67-AGR-022).

## Field Study Permissions

The following information was supplied relating to field study approvals (i.e., approving body and any reference numbers):

Kasetsart University.

## Data Availability

The raw data are available in the Supplemental File.

## Supplemental Information

Supplemental information for this article can be found online at http://dx.doi.org/10.7717/peerj.19190#supplemental-information.

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
