# Peer review of "Comparative larval morphology of four Pteroptyx (Coleoptera, Lampyridae, Luciolinae) species in Thailand"

_PeerJ, doi:10.7717/peerj.19190_

## Round 0.1 · original submission · Major Revisions

Please, address all the reviewers' comments. Notice that there are also annotated files attached to the review.

·

Basic reporting

BASIC REPORTING. This is a well-structured document that reads well for the most part and is clearly expressed. However, I would like to see many situations readdressed so we can see this necessary and useful contribution to Lampyridae taxonomy available in published form soon. The introduction attempts too much – and this is not evidenced in the outcomes in the paper (1 below). Literature coverage is inadequate and has affected how they have expressed findings (see 4, 5, 6, 11, 12). An overview of the body plan for Pteroptyx larvae is suggested; there seems to be a tacit assumption that the reader will determine this from references given; presently there is no way for the reader to interpret the ventral body plan (5). Descriptions are headed Diagnosis but attempt to be both a description and a diagnosis which is confusing; suggest they be split into diagnosis and description as separate sections (8). Dichotomous key can be improved (10). Figures for the most part are good and clear but scale lines need attention on some (13); text and figures do not always agree (7, 13), and figure legends should be self-standing (13).
Reviewer comments summary:
The following comments summarise my review: I also provide a detailed breakdown for the authors.
Pleased to see this work commencing on larval identification and look forward to seeing it soon in print. The comments below are designed to help you improve your work. Do not let negative comments discourage you, this work needs to be published.
BASIC REPORTING. This is a well-structured document that reads well for the most part and is clearly expressed. However, I would like to see many situations readdressed so we can see this necessary and useful contribution to Lampyridae taxonomy available in published form soon. The introduction attempts too much – and this is not evidenced in the outcomes in the paper (1 below). Literature coverage is inadequate and has affected how they have expressed findings (see 4, 5, 6, 11, 12). An overview of the body plan for Pteroptyx larvae is suggested; there seems to be a tacit assumption that the reader will determine this from references given; presently there is no way for the reader to interpret the ventral body plan (5). Descriptions are headed Diagnosis but attempt to be both a description and a diagnosis which is confusing; suggest they be split into diagnosis and description as separate sections (8). Dichotomous key can be improved (10). Figures for the most part are good and clear but scale lines need attention on some (13); text and figures do not always agree (7, 13), and figure legends should be self-standing (13).
DESIGN. Adequate. Suggest provide more information about the original habitats. Some procedures should be expanded (2, 3).
VALIDITY OF FINDINGS. Outcomes are not consistent with what was promised in the introduction; this relates especially to the inadequate literature coverage which could have aided them (1); some items need to be expanded (2) or given more precise descriptions (3); the actual taxonomic descriptions make no reference to fully described larval characters in references listed below (and for which a scoring matrix is available that would permit them to make comparisons with a wide range of larvae) (5, 12); there is no indication of how they determined what instar they were dealing with (2); Comparison with other Luciolinae larvae (not described in anywhere such a level of detail but still useful) could give them a better perspective (4, 6, 8, 12); some characteristics need to be reassessed as they are incorrect or poorly described (6, 7).
1. Unsubstantiated statements – page 1, lines 25-27, 35-38. Much is made about habitat restoration but this paper does not provide that information.
2. Rearing – line 95. Needs expanded procedures; there is no distinction of instars nor information on how they might have told which instar is which.
3. Measurements – lines 123, 130. Need precise descriptions on how to obtain these; no indication on how they measured the whole body length.
4. Taxonomic descriptions line 138. Need to consult other references which list larval characters and have used them in phylogenetic analyses (e.g. Ballantyne et al 2015, 2022) as a basis for comparison.
5. General characteristics Pteroptyx larvae – line 155. Needs a general overall description to put reader in the picture (appropriate references suggested here); ventral surface of Luciolinae difficult to interpret otherwise.
6. Characteristics incorrectly interpreted, poorly described or not addressed at all – lines 162, 197, 240 colour described as fade, lighted or dark markings – there are not colours; lines 172-177 some of the pictures do not match the descriptions of the antennae, lines 183-190 do not match palpi, line 190 mentions a sensorium but there is no label on figures and difficult to see what they mean, 190, 195 are simply unclear (especially texture of dorsal surface), 200 is incorrect ( leg segments mislabelled), line 370 too much subjectivity in description of dorsal texture, line 372-380 position of mesothoracic spiracles, line 232 you need to define where the episternum and epimeron are,
7. Errors - lines 172-177 some of the pictures do not match the descriptions of the antennae, lines 183-190 do not match palpi, line 190 mentions a sensorium but there is no label on figures and difficult to see what they mean, 190, 195 are simply unclear (especially texture of dorsal surface), 200 is incorrect (leg segments mislabelled)
8. Diagnoses/Descriptions from line 155 – for all the species addressed. For each species a section labelled diagnosis is also the species description and they need to be separated. The description should address all the features you describe. The diagnosis only addresses those features you think will give adequate separation from the other Pteroptyx species you address (include maipo here) and look at your key – if you have constructed it appropriately this is where you find your diagnosis for each species.
9. It follows that descriptions are often inadequate – lines 218, 228, 232
10. Dichotomous key – line 317 is not strictly dichotomous and needs to be modified.
11. Disagreement – lines 360, 435 usually arise because of inadequate literature coverage.
12. Literature coverage is inadequate. Ballantyne et al 2015 listed larval characters including some they use here, described them and species were scored and used in a phylogenetic analysis (matrix data available allows for comparison of many species). Larvae with similar morphology to Pteroptyx (but with much less attention to detail) addressed in Ballantyne & Lambkin 2000. They should not just concentrate on Pteroptyx but explore all Luciolinae larvae more fully.
13. Figures nicely done for the most part but do not always corroborate what the text is saying; scale lines on several need to be checked; figure legends should stand by themselves and not require reference back to somewhere in the text.

Experimental design

DESIGN. Adequate. Suggest provide more information about the original habitats. Some procedures should be expanded (2, 3). Most is covered in the section above.
Please see attached document reviewer comments summary 20 June

Validity of the findings

VALIDITY OF FINDINGS. Outcomes are not consistent with what was promised in the introduction; this relates especially to the inadequate literature coverage which could have aided them (1); some items need to be expanded (2) or given more precise descriptions (3); the actual taxonomic descriptions make no reference to fully described larval characters in references listed below (and for which a scoring matrix is available that would permit them to make comparisons with a wide range of larvae) (5, 12); there is no indication of how they determined what instar they were dealing with (2); Comparison with other Luciolinae larvae (not described in anywhere such a level of detail but still useful) could give them a better perspective (4, 6, 8, 12); some characteristics need to be reassessed as they are incorrect or poorly described (6, 7).

Additional comments

The following is a detailed page by page indication of areas that need attention:

Well done for a first attempt but it needs much more work. Good to see you are investigating larvae, an area much in need of detailed work as you have done here. Keep it up.

Page and line number Issue Comment
Page 1 Background and Discussion
You make two statements about habitat
Identifying species at the larval stage is crucial for habitat restoration
And accurate identification is crucial for understanding their microhabitats. Be careful here you are overselling what you have done. You have not been able to identify the habitats from what you present here.
Perhaps you could include a little information about the nature of the habitat where you collected your specimens as that would be very useful.
Paged 1 Results Pronotum shape Protergum shape (usually restrict use of notum to actual wing bearing segments)
Line 24 worldwide Really? Worldwide? Perhaps rewrite
Lines 25-27 Crucial for habitat restoration etc Same comment as above; you may be able to use this statement if you reword it carefully.
Line 35, 36 Likely reflecting their specific requirements No this is not what you show. You show larvae that are very similar to other described Luciolinae larvae (which I will point out later on) and only in P. valida have you found something quite distinctive, and that is only in comparison with three other larvae. There are other Luciolinae larvae with dorsal tubercles that you do not address. Ballantyne & Lambkin 2000 addressed very similar larvae with dorsal tubercles that you could compare with.
37, 38 Accurate identification …crucial for ..microhabitats Is it really though? You are not yet showing that in this paper
Line 61 The lack of knowledge…which directly impact firefly populations Suggest rewrite this sentence it does not make good sense as written
Lines 77, 78 As only two species of Pteroptyx thus far have well described larvae we clearly need much more
Line 95 rearing I assume you reared your larvae through from egg to adult as you mention food throughout all larval stages. If this is so and you have clearly been successful then you need to expand this section – rearing larvae is difficult and others can benefit from your expertise.
I also assume that you have different instars which you could identify and differentiate? You do not seem to have mentioned how you know which instar was which.
Lines 101-103 As above how did you know which instar was which? Any measurements?
Line 123 measurements I think you should be quite precise in your explanation of just where you are making these measurements especially that of the length and width of the Protergum; there is a lot of margin for error here. Maybe specify within the coloured area, or between the tubercles?
Line 130 Length and width of whole body How did you do this and not be subjective??
Line 138 taxonomic description There are other references appropriate here. Ballantyne et al. 2015 scored larvae using 48 characters some of which are similar to yours. See also Ballantyne et al. 2022 for more larval characters.
Line 155 general characteristics If you look at Ballantyne et al 2022:24 they give an outline of the basic design of Luciolinae larvae (with modifications of course as they were addressing specifically aquatic larvae). Your section could be improved if you did the same so your reader understands what they're seeing. You do not state that the overall body plan is quite similar to what had already been described and this leaves one wondering how many of those features mentioned in the other references apply here. If you were to apply what is outlined in Ballantyne et al. 2022 page 24 then your readers will get here a complete picture of what these Pteroptyx larvae look like. This relates especially to the ventral surface (median sternum, laterosternites and laterotergites in the abdomen), the structure of the thoracic segments 2 and 3 from below, and also the basic head structure. Yes it is just repetition but it needs to be said.
Line 155 Diagnosis? This section can be much improved by indicating how Pteroptyx larvae differ from what is already known of other Luciolinae larvae. You are not telling us diagnostic features here if you do not mention other genera too (you are just distinguishing your four larvae from each other). It is very probable that larvae of a whole group of Luciolinae – Pteroptyx, Colophotia, Australoluciola, Pyrophanes, Medeopteryx etc are all of the same form. Make your point that we know very little about them (but we do know something!!) and this will enhance what your paper is about
Line 162 and elsewhere. Colour descriptions I think colour is inappropriate here as it seems to be quite variable anyway.
Line 169 Picture references A good clear shot of the dorsal and ventral head with mouthparts intact is needed. The anterior margin of the dorsal head between the antennae often has a distinctive shape and I can’t tell from what you present.
Lines 172-177 Description is inaccurate and does not match your figures The antennae can be retracted into a long articulating membrane which you don’t mention; the basal segment is short, segment 2 is as wide as the basal one but much longer; segment 3 is tiny by comparison in both length and width and have apical setae; the sense cone in this genus has characteristic dimensions – look at its height relative to segment 3 and its width also.
Lines 183-190 See how you might describe the head with a fused maxillolabial complex Description of palpi here of maxilla does not match your figures – do you recognise a palpifer or a palpiger? The very largest segment at the base of the maxillary palp is usually referred to as a palpifer.
Line 190 sensorium I could find no obvious sensorium in any of these pictures. Fu et al found in terrestrial species that such a structure was apical but you don’t seem to say where it is.
Line 195 Texture of tergites Possible to either describe this better or show us in a picture/
Line 197 Colour Just say they were variable? Or is there some way to describe this to contrast with what you might see in other genera? It is a dull colour pattern isn’t it?
Line 200 Legs Legs are 4 segmented coxa trochanter, femur and tibiotarsus.
Line 205 and following Expand on just what the structure of the abdomen is – you leave too much unsaid; see previous comments
Line 205 abdomen What about segment 10? You can see the outline in your figures
Each species treatment You give a diagnosis which is also your description of the species concerned. This is unsatisfactory. The diagnosis has to distinguish the species from any other Pteroptyx we know about. The description needs to be separate and may be repetitive, as you need to show what the specific features are of the species concerned (remember you have already covered what you think are generic features in your first section).
In describing the tubercles can you say how they differ from one species to another?
Don’t say things like distinguished from …by the shape of protergite and colour pattern on ventral side and leave it up to the reader to figure out what you mean, you spell out exactly what those differences are. You have done this quite well in some areas and not as well in others. Split each section into two parts – short diagnosis so we can see just how they differ, and an expanded (if necessary) description which will include those features as well.

Might it not be just as important to give a diagnostic section for the first instar larvae as well?
218 diagnosis and subsequent diagnoses Distinguish each species from all known Pteroptyx species so include maipo too.
Line 218 diagnosis Read this again – you can be quite specific about where the tubercles are – anterior, lateral, across posterior margin, at the posterior corners etc.
Line 228 Fade diamond shape Faded? Be more specific. Is it a colour?
Line 230 where located a pair of light organs on ventral side rewrite
Line 232 Episterna epimera Have you identified these areas previously? How are we to know what they are?
Line 237 Great that you try to differentiate the first stage larvae. Can you devise a key for them as well? When do they approach the same form as the fifth stage larva?
Line 239 tubercles on all tergites formed except on
protergite So protergite does not have any tubercles developed yet in this first stage larva?
Line 240 which are pale on anterior and median region Pale is not a colour
Line 241 uniformly fine granular distribute on distributed
I did not go through the rest of the descriptions but you should be able to follow what I have already indicated above and make some changes for yourself. Your descriptions of these larvae are your diagnoses. Can you not separate a diagnosis) which can be quite short) and then give an expanded description/redescription?
Line 317 Key to species A dichotomous key has to have similar contrasting statements in each couplet. Your couplet 1 contrasts tergites smooth or rough, and then does not give any information about the pronotal length width or the colour of the ventral side. Check your key to ensure you correct this.
Line 335 and following How significant do you think any of these characters you mention here will be for people collecting in the wild? Or will that even be an issue? Will they identify in the laboratory?
Line 360 Templates I disagree with your interpretation here – larval morphology has been expanded as more and more larvae were found and many Luciolinae larvae have now been scored from morphological characteristics which show they are of as much use as the adult features in estimating relationships. The basic format for describing larvae started long before any Pteroptyx were described
Line 368 this prior assemblage of characters is not useful for species Don’t criticise – use this to show what you have achieved – you now can use more features as a result of this paper
Line 370 the character was not found to be variable across But the problem is in how you define it here; I am not sure if by rough you mean having tubercles (which is enough in itself) or there is another feature which you have not described well nor illustrated. If it is important try to describe it better
Line 372 Position of spiracles Somewhat subjective interpretation here, can you give us some pictures?
Did you attempt to determine if they were all the same type of spiracle?
Line 383 intraspecific variation in the color pattern of the prosternum was examined Reword this – you determined there was intraspecific variation
Line 386 morphological variation in Pt. malaccae adults was found in different What significance is this here where you are discussing larvae/ Did you establish any variation in the larvae?
Line 424 Larval mouthparts What you see here is typical of this family
Line 426 Retinaculum with teeth (it can have one or two) In Pygoluciola the larvae were semiaquatic – difficult to hang on to prey? Fu described Pygo qingyu which attacked ants in a head to head combat – could the toothed mandibles help in such a situation? There are other examples of mandibles with teeth in terrestrial species (look at Abscondita).
Line 435 The sense cone of the antennae might be important for I disagree with lines 435-437. Shape of the sense cone can vary (short and flat versus long and thin.
There are references you can assess that describe the sense cone in aquatic species so you can reassess your statements here. Check your literature and look at Ballantyne et al 2015 data matrix where sense cones' length and width were scored for many species
Line 352 Discussion You have made a good job here of trying to evaluate your results.
Line 354 references Ballantyne & McLean 1970 was the first modern day Pteroptyx treatment.
Line 360 morphological descriptions of Pteroptyx larvae have been used as templates Not so – check the other references I have given you previously. There is a basic plan to the Luciolinae larvae which we have gradually discovered the more larvae we see.
Line 368 this prior assemblage of characters is not useful for species dentification in Pteroptyx But this assemblage of characters is useful to define the genus.
Lines 372-380 Position of mesothoracic spiracles Somewhat subjective interpretation here can be improved with another set of pictures
Line 390 measurements See previous comments; you need to specify just how you measured the entire length
Line 396 larva size will be irrelevant It might be irrelevant we don’t know yet
Line 400 There were variations in the morphometrics of species observed under the microscope Which species are you referring to here? Can you give a reference? Be more specific here.
Line 424 We assume this is also true for Pteroptyx species. What has been done so far on Luciolinae mouthparts shows a remarkable uniformity of structure, with the variability being in the mandibles (are they toothed or not). We already know how these mandibles work – they are modified for injection of midgut juices to paralyse the prey – perhaps some more references here?
Line 435 The sense cone of the antennae might be important You seem to be unaware of Ballantyne et al 2016 Zootaxa 3959 which listed many larval characters and scored larvae and included them in the phylogenetic analysis (look at figures 2b and C). The sense cone was addressed in characters 371, 372 and the complete scoring matrix would allow you to make comparisons between various larvae used here
Line 438 except for size of the sense cone of the antennae, which might indicate a different feeding habit? You do not actually describe the sense cone for each species– are we expected to look at your figures and determine this ourselves? In Table 2 despite all the material you examined you can give only one ratio????? No range?
Figures Nice figures well done
CHECK ALL YOUR SCALE LINES Where you have extra indications on the figures include that in the figure legend, they have to stand by themselves. Example figures 1, 3, figure 9 is this dorsal or ventral?
Figure 24 needs better focus
Not all figures show what you are describing (see above)

Reviewer 2 ·

Basic reporting

The text in the manuscript is clear and written in professional English. The Introduction could provide more information about larvae within Lampyridae and the subfamily Luciolinae to guide the audience on the topic. Since the topic is about the comparative morphology of firefly larvae, it would be interesting to know the current state of knowledge in larval morphological studies in general, and specifically in Luciolinae. Is it only Pteroptyx, or do other genera also lack information? Lines 48–54 could be shortened to emphasize that conservation, while important, is not directly measured in this study. Classic literature (Buck, 1938) could be replaced with more recent studies, unless Buck's work contains specific information not found elsewhere. The figures are relevant and high quality, but their labels and descriptions need improvement.

Experimental design

Original primary research is within the scope of journal. The objective is vague. The authors should indicate precisely that they only examined, documented and compared the larval characters of four Pteroptyx species in Thailand which discussed the limitation of the study. Methods are generally in order, but some aspects of taxonomy can be expanded. Perhaps a table of the abbreviations of taxonomic characters would be helpful to refer to when needed quickly.

Validity of the findings

• Out of 41, please describe how many larval specimens by each species were examined.
• It isn't clear to me how many instars have been examined (only the first instar?), but the main reference quoted the last instar from B&M 2002. I am also wondering how you determined the first and last instar. Are there any differences in the morphological characters of these specimens within each species?
• Lines 147-153, spell out the genus name for each species. Scientific name should include a comma between author and year
• In your description of each morphological character, you should be able to link each figure to each morphological description, which will be helpful for readers. The description of every species should be expanded, but the description of the larval genus can be summarised.
• Consider modifying the scatter plot, as the labelling is quite confusing. See comments in the reviewed PDF. a, b, c labels are quite close to each other, which can create confusion. Consider moving a to the upper left corner or making space in between graphs. Check spelling error for Pt. asymmetria.
• It isn't clear if the authors re-examine the specimens or only quoted B&M: "The larval characters of the first to third, as well as last instar larvae were comprehensively described based on reared specimen in (Ballantyne & Menayah 2002)." I suggest rephrasing this statement.
• While the figures are great, they lack clear descriptions of each character. For instance, Line 263 SC= 23A3 (Fig. 26) doesn't interpret much, and when I referred to Fig. 26, there isn't any label of SC. Please check throughout the texts. FIg. 9 What's in the box? Describe it in your caption. Label each segment if possible. Include these abdominal segment numbers in your caption
• Discussion should include the limitations of the study and what can be improved. For instance, how does this study address the issue of geographically structured populations of certain Pteroptyx species? Can larval species-specific traits be distinguished in cryptic speciation?

Additional comments

The study provides valuable insights into the larval morphology of the Pteroptyx genus, particularly in the context of the limited existing reports on this topic. The authors examined four Pteroptyx species in Thailand using species-specific traits that could significantly contribute to this field of taxonomy. However, several improvements should be addressed prior to considering this manuscript for publication in PeerJ.

Annotated reviews are not available for download in order to protect the identity of reviewers who chose to remain anonymous.

Reviewer 3 ·

Basic reporting

no comment

Experimental design

no comment

Validity of the findings

no comment

Additional comments

This is overall a well-written study, and the introduction does a great job about increasing awareness of firefly conservation and strategies to continuously preserve those endangered populations. I think the title of the paper could be changed to reflect the ideas of the authors to use those fireflies as instruments of biodiversity conservation. The morphology of the work is fine, but I think the discussion was missing on some key elements about how the author's studies will contribute to the conservation efforts of fireflies, and particularly of Pteroptyx.

I attempted to minimize the use of superlatives and other subjective expressions (e.g., Unfortunately, amazing, fascinating, impressive, etc.) and made other comments directly in the manuscript. It was very inconvenient to review the images provided, because authors only provided eps files, instead of simple jpegs or pdfs.

I would like to see what were each author's contributions to this paper.

Annotated reviews are not available for download in order to protect the identity of reviewers who chose to remain anonymous.

---

## Round 0.2 · Major Revisions

Please, address all the reviewers' concerns and suggestions. Note that there are many technical issues that were missed in your revised version.

·

Basic reporting

Please see detailed comments in section 4. Additional comments below where suggestions for amendments or improvements are made.

I am responding to their response to my first review of this document. I am not just doing here another evaluation and possible acceptance of their responses to the issues I raised in my first review, but am having to do another review
All of my comments are detailed in the section 4. Additional comments below.
This is not yet satisfactory and indeed much of their attempts to improve it (with additional material) have only added problems which I will outline below. Most of my comments below (in section 4) refer to the additional sections still in red or blue (indicating tracked changes), and I indicate how these can be improved.
One of the criteria for publication in this journal appears to be addressing one of the global challenges we are facing. I think it is trying to achieve this requirement that most of the problems have arisen. My major problem relates to what they actually do here (describe characters of four larvae and reliably identify them) for the first time (and an important contribution), versus much else about the importance of habitat restoration which is not directly dealt with in this paper.
The abstract is of a form I am unfamiliar with and does not state concisely the results of this exercise.
The english in places needs improvement, some spelling mistakes and some sections could be made much more concise.
Agreed changes (from response to review document) to terminology have not been universally accepted.
Literature references have been improved but references are not in strict alphabetical order.
Figures and figure references are detailed below and require some amendments.
Results here are descriptions that differentiate between four different Pteroptyx larvae.

Experimental design

Please see detailed comments in section 4. Additional comments below where suggestions for amendments or improvements are made.


One of the criteria for publication in this journal appears to be addressing one of the global challenges we are facing. I think it is trying to achieve this requirement that most of the problems have arisen.

I do not feel that they achieve that in this paper. I cannot stress this too strongly – they have identified, by breeding through and then comparison of bred males with males of already identified species, four species of larvae of which one had already been described though not in such detail. Their Methods mention nothing about habitat assessment.

This is an important contribution in itself.

I surmise that some of their problem is that they have identified factors affecting larval habitat (they state in their response letter “results useful for further studies” and “microhabitats were analysed – explained in a future publication”) but these were not part of this paper.

However, they make a great deal of how this could be used in this paper in determining habitat requirements (true) but this is not what they do here and I find this an unacceptable example of overpadding the content. I suggested below that this information could be better included in their discussion and clearly the following as yet unpublished paper which I feel will answer most of my concerns.

Validity of the findings

Please see detailed comments in section 4. Additional comments below where suggestions for amendments or improvements are made.


These are important findings as we know so little about differentiating features of Luciolinae larvae, and what they have achieved will be useful.

Additional comments

For my comments below I have used page and line numbers consistent with the final Word document having track changes, unless I indicated otherwise.

Abstract.
1. I am not familiar with an abstract that seeks to address so many issues (background, methods etc). However shouldn’t an abstract still present in a succinct form just what the outcomes were? This does not do that.
2. Introduction lines 29-35; good points but this is not what you have done; you can use this but rewrite it to reflect what you actually did.
3. Lines 36, 37 you examined characters using a microscope; you identified the larvae by breeding them through to adult and comparing the adult males with already identified male specimens.
4. Lines 48, 49 – “likely reflecting…” but you do not show this either. Move to your discussion.
5. Lines 51-53 yes we all agree with these sentiments, but this is not what you did. Perhaps move this to your discussion and make all these useful points there

Introduction.
1. I like how you have tried to set the scene here. Check Josoh on line 73 you mean Wan Jusoh I think.
2. The second and third paragraphs lines 71-109 really need to be shortened. You need to stress that to begin one of these conservation programs you need to be able to identify the larvae; this is the absolutely necessary first step and has proved difficult in the past and that is what you are doing here.

Methods.
1. Morphological examination. lines 139- 142 why are there two sentences here; were there two different procedures to kill and fix the larvae? Remember you are killing, fixing and preserving them by immersion in ethanol.
2. Lines 142, 143 I can’t follow this and it is a critical part of your paper – being able to identify just what instar is what. You mounted them once they had moulted but what about them then told you which instar they were? Just because you could compare with a labelled slide? You need to be clearer here. State clearly if you couldn’t identify them.
3. Line 188. You wanted to achieve a horizontal inclination so you can measure and photograph the prothorax and the protergum. A neat solution but there are much easier ways to do it. But I like the use of the ice!! Sensible solution. However if the prothorax is always a little “drooped” then using that feature might be difficult for anyone looking at fresh larvae in the field too wouldn’t it?

Taxonomic description
1. You used the characters outlined in these papers? At some stage then it would be appropriate to say how you have expanded on them or if they were useful and what else you needed to do. Should be in methods.

Results

1. Generic diagnosis. Some features here are not diagnostic but are common to all the larvae we know about – antennal segment 3 is much shorter than the other antennal segments and they all have a sense cone (its size and shape varies). A reference will allow you to confirm that (Ballantyne et al. 2015 Zootaxa 3959 has list of characters and many species including larvae scored using those characters). You are distinguishing the first stage larvae from the older larvae here but you need to say what is it about the texture of the tergites and what about the colour on the ventral side is distinctive. All Luciolinae larvae that have been described (except Sclerotia) seem to have the same ventral body plan as Pteroptyx. What definition of Colophotia are you referring to (needs a reference here).
2. Be clear that you are only using the features of your four species as you do not give any extra information about the larvae of P. maipo in your generic description.
3. You seem to have struggled with giving a generic description based on your four species and then having to write separate individual descriptions. Try to say what is common to them all and then add if something is variable. For example around lines 296 on you could simply say that colour patterns differed and then you can expand exactly how they did in your species treatment. Apply this more widely there are many places where this would improve your expression.
4. Lines 242 and 247 conflict as I interpret sclerotization as hardening – membranous and very soft bodied versus dorsal plates heavily sclerotised; might help to indicate in your methods how you interpret sclerotization – is it just dark pigmented areas or are you saying that dark areas are also sclerotised; do you then infer that sclerotization is also hardening of the cuticle?
5. Lines 248, 251 are incorrect. There are 9 obvious abdominal segments (your line 239; your diagrams label abdominal segments 1 – IX); there is no abdominal tergite 11 (Line 248).
6. Lines 255, 256. Interpretation of these areas changed in Ballantyne et al 2022 (presternum to an anterior basisternum and sternum to sternellum) and you need to indicate which version you want to follow. Put it in your methods section.
7. Line 257-258 can’t follow this? You mean these areas can be pigmented or not pigmented? But you need to define these areas too.
8. You need to add here what the structure of the abdominal segments is – median sternal area with laterosternites at the sides, and paired laterotergites with spiracles above that just beneath the terga.
9. Line 265 around and behind what? Line 274 compareD to
10. Line 279 one long seta parallel (to what?) in median region of ventral and dorsal inner side – not sure what you mean – maybe put some labels on your figures
11. Line 282 how many segments are you giving the labial palpi? What about the postmentum? It is the rest of the ventral head area between the maxillae.
12. Line 284 cardo – no figures show the cardo and I think you are confusing it with the larger area which is the stipes.
13. Line 286 with conical what?
14. Line 287 see discussion elsewhere about palpifers
15. Line 289 what does subcorneal mean? There is no obvious sensory area anywhere on the apical palpomere?? In some aquatics it is along the surface behind the apex
16. Lines 295, 296 good idea to describe more what you mean by the rough texture as the figures do not convey it well. Did it have very short spines for example?
17. Lines 303, 304 recommended leg segments be referred to coxa, trochanter, femur and tibiotarsus (this has the apical claw). Terminology here is confusing, tibiae and tibiotarsus; you need to modify this on figure legends if you accept this terminology or state in your methods what version you are following and then be consistent. You can make your own case.
18. It is the prothorax that contains the retracted head WITHIN it not beneath it.
19. Line 319 P. maipo has nodules along the posterior margins of the terga (you say tubercles along the posterior margins of most terga; Ballantyne et al 2000 figure 61). They are the same thing.
20. Unfortunately this now means that all your diagnostic sections are incorrect when you refer to P. maipo as not having tubercles along the tergal margins and will have to be amended.
21. I did not go through any more of your species descriptions, but I suggest that If P. maipo actually did have tubercles/nodules along the posterior margin of the terga then you should look very closely at what you are calling valida; these two species are very similar and we might expect their larvae to be similar too. In your pictures there appear to be some sort of elevated areas along the posterior tergal margin, perhaps you could have another look and see if you can define it better. I can’t tell very well from your picture

Terminology.
The following have not been amended throughout or are inconsistently used:
1. You agreed that leg segments would be referred to as coxa, trochanter, femur and tibiotarsus. Table 2 still refers to a tarsungulus; tibia has two mentions in the version with track changes accepted.
2. You agreed to use protergum instead of pronotum for the first body segment dorsal plate; pronotum still appears 14 times in the version with track changes accepted.
3. In lines 189, 191 there is some confusion over the use of pronotum and I think you are using this as equivalent to prothorax and it should be changed or its use clarified.
4. The use of palpifer and palpiger to refer to what appears to be a large basal segment of the maxillary (fer) or labial (ger) palps is common to those studying Coleoptera (see Lawrence and Ślipiński (2013) Australian Beetles I: 81 for larvae. I mentioned it as numbering the palpomeres may cause confusion (are there 3 or 4 in the maxillary palp for example).
5. Colour patterns need some improvement and should be a simple exercise as light and dark are not colours (yes I was chided for this myself by a reviewer too). Amend so you use the degree (dark or light) with an actual colour.


References
Your references are not always in alphabetical order but chronological order of the first author. I am assuming this journal follows a conventional line here.

Figures
1. Some are not labelled correctly as they are not all showing the entire area like all of the ventral head area; if you are only showing part of an area then you should specify that.
2. None of your pictures show the cardo, which is the small area at the base of the maxilla; as I interpret it I think the larger area you refer to as the cardo is actually the stipes.
3. Fig. 12 for example shows the hypopharynx but you don’t mention it.
4. You don’t have any good clear shots of the dorsal head but it could be along the anterior margin of the frontoclypeus that there will be distinctive outlines and I simply can’t see that area clearly on any of

Reviewer 2 ·

Basic reporting

The authors have made some revisions to the text, which is appreciated. However, addressing the previous comments in more detail would be beneficial. For instance, instead of providing context on firefly larvae, authors narrowed the scope to only Pteroptyx. Additionally, some paragraphs contain multiple topics, leading to a lack of clarity. It might help to create clearer connections between sentences to enhance the flow and coherence of the text. Lines 58-66, for example, shift from the hypothesis on synchronous flashing to the economic impact and decline in population due to destroyed habitats.

Experimental design

The objective is still vague. In lines 91-93, the authors indicated, "Although the larval morphology of these two species was well described, knowledge of the larvae from the remaining Pteroptyx species is sorely needed." However, they didn't elaborate on what knowledge gap needs to be filled.

In the 'rearing' section, they didn't indicate how many adults and larvae of Pteroptyx spp. and what species are involved. They have added a table of the abbreviations of taxonomic characters but also included the abbreviation of a non-taxonomic character (see line 151).

In the 'taxonomic description', lines 182-183, the authors state, "The larval characteristics of each species largely correspond to those described for P. valida larvae by Ballantyne & Menayah (2002)," but this wording is unclear.

Validity of the findings

Please be consistent in the citation style of taxa authors. The authors should adhere to zoological nomenclature as per instruction (Linnean binomials). Authors of taxa are indicated with a comma, e.g. Pteroptyx tener E. Olivier, 1907. Sometimes the author's name has no initials, sometimes it has. See line 291. asymmetria is not a new combination, the taxon author and year shouldn't be in parenthesis.

Description/Redescription of larvae. It is unclear why only selected instars are described instead of all instars. The flow of writing could be better organized. In line 358, a description of the sixth instar is presented first, followed by the first instar. Why isn't it the other way around? How you determine the distinction between the 4th/5th instar and the 6th is also unclear. Please see line 391, where the measurement ranges from 8 mm to 12 mm (a 4 mm difference). The key to species using larvae is confusing. A ventral surface with dark pigmentation isn't a helpful character, especially when there's a possibility of a slight degree of variation between species in terms of colouration.

Fig. 64. The scatter plots do not explain what the small letters "ab" and "b" represent.

Additional comments

I appreciate the modifications made by the authors. However, several areas still need to be strengthened for the manuscript to reach its full potential. My comments mostly cover the technical aspects of writing, not taxonomy, as I trust that will be addressed by another reviewer.

While there is valuable information presented, the analysis on conducting comparative studies could have been done better. The findings would be more impactful with clearer presentation and organization. The authors need to be consistent in how each larval stage is studied, which will enhance the overall clarity.

If I were someone working in the field with different kinds of larvae, I would want straightforward guidance on determining species based on morphology. For instance, an identification chart for larvae would greatly contribute to the field and enhance larval identification practices. Otherwise, researchers might resort to using DNA sequencing linked to larval-adult associations because it is much faster (and reliable) in terms of species identification.

I appreciate the authors' efforts in developing this manuscript. However, it is not yet suitable for publication in its current form. With a few revisions for better organization and coherence, I believe it could make a significant contribution to PeerJ and firefly research community.

Reviewer 3 ·

Basic reporting

The authors extensively addressed to all reviewers comments and suggestions, I recommend the publication of this manuscript in its current form.

Experimental design

N/A

Validity of the findings

N/A

Additional comments

N/A

---

## Round 0.3 · Minor Revisions

Please, address the reviewer's final comments and suggestions and carefully check for potential mistakes.

Reviewer 2 ·

Basic reporting

The authors have significantly revised the text, which is appreciated. It has improved greatly.

Minor suggestion. Lines 88-90 can be modified slightly to improve clarity: "We compare the morphology of instar larvae and provide a larval identification key for four Pteroptyx species currently known from Thailand". The details of the instar can be elaborated in the Method section.

Experimental design

Lines 116: These four species are mentioned for the first time in the text and should be spelled out in full.

Lines 116-117: My understanding is that this study focused on examining first instar larvae for all species. Then, it looked at 5th instar larvae for the P. asymmetria and P. malaccae, while P. tener and P. valida were assessed at the 6th instar stage. I recommend revising the sentence to improve the clarity.

Line 125: "At least three larvae of each species" - please clarify if these are taken from the 12 individuals or if there are additional specimens.

Validity of the findings

Line 194: Add 's' to Fig.

Line 221: "Figs. 55-60" refers to figures from other studies and are not part of this study. Please consult the journal format or check with the Editor regarding the best way to cite figures from different sources to avoid any confusion with figures from this study. Typically, we use small capital letters to indicate that the figures are sourced from elsewhere.

Line 301: once again, Pteroptyx asymmetria isn't a new combination. There shouldn't be a parenthesis around the taxon author and year. Correction: Pteroptyx asymmetria Ballantyne, 2001

Line 332: There must be a comma after Gorham. Please be consistent throughout the manuscript. Correction: Pteroptyx malaccae (Gorham, 1880).

Line 360: Same comment as above.

Line 373-374: Is it Type One or Type 1? (see line 340). Please be consistent. This applies to Type Two. You refer to Fig. 5c and 5e but I can see any mention of Type One/1 or Type Two.

Line 387: Pteroptyx valida shouldn't have parentheses. Please remove.

Line 437: Remove "." after Table 2. Please check this typo throughout the manuscript.

The authors provided a justification for using certain species as either fifth or sixth instars in their rebuttal response; however, this explanation was not included in the main text. I believe it would be beneficial to add this disclaimer so that readers can understand the reasoning behind this choice, especially as they may seek to replicate the study.

Additional comments

Unfortunately, I cannot view Supplementary files, especially their figures, because they are in EPS format, which I don't have access to. It would be helpful to use a common file format, such as PDF or JPEG/PNG, for reviewing purposes.

---

## Round 0.4 · accepted · Accept

I think the reviewers' comments were addressed and the manuscript is ready for publication.